# When Model Knowledge meets Diffusion Model: Diffusion-assisted Data-free Image Synthesis with Alignment of Domain and Class

Yujin Kim [* 1 2]  Hyunsoo Kim [* 1 2]  Hyunwoo J. Kim [3]  Suhyun Kim [4]

## Abstract

Open-source pre-trained models hold great potential for diverse applications, but their utility declines when their training data is unavailable. Data-Free Image Synthesis (DFIS) aims to generate images that approximate the learned data distribution of a pre-trained model without accessing the original data. However, existing DFIS methods produce samples that deviate from the training data distribution due to the lack of prior knowledge about natural images. To overcome this limitation, we propose *DDIS*, the first **D**iffusion-assisted **D**ata-free **I**mage **S**ynthesis method that leverages a text-to-image diffusion model as a powerful image prior, improving synthetic image quality. DDIS extracts knowledge about the learned distribution from the given model and uses it to guide the diffusion model, enabling the generation of images that accurately align with the training data distribution. To achieve this, we introduce *Domain Alignment Guidance* (DAG) that aligns the synthetic data domain with the training data domain during the diffusion sampling process. Furthermore, we optimize a single *Class Alignment Token* (CAT) embedding to effectively capture class-specific attributes in the training dataset. Experiments on PACS and ImageNet demonstrate that DDIS outperforms prior DFIS methods by generating samples that better reflect the training data distribution, achieving SOTA performance in data-free applications.

## 1. Introduction

The widespread availability of open-source pre-trained models have significantly advanced the field of deep learning (Ridnik et al., 2021; Singh et al., 2023; Goldblum et al., 2024). Recently, platforms such as Hugging Face (Wolf, 2019) have further enhanced accessibility of the pre-trained model, enabling researchers to easily apply these models to various tasks like knowledge distillation and model pruning (Tran et al., 2024; Wang et al., 2024; Yin et al., 2020). However, a common requirement for utilizing these models in such applications is access to their training data, either in full or in part, for training student networks or fine-tuning pruned models. Unfortunately, these training datasets are often inaccessible due to various reasons, including data privacy and copyright issues. Consequently, the lack of access to training data presents a challenge in leveraging the potential of pre-trained models across various applications.

To address the above issue, *Data-Free Image Synthesis* (DFIS) has been proposed as a solution. DFIS aims to synthesize images that approximate the training data distribution by extracting the model's internal understanding of its training data. This enables the application of these models to tasks like data-free knowledge distillation or pruning, enhancing their utility (Yin et al., 2020; Mordvintsev et al., 2015; Kim et al., 2022; Ghiasi et al., 2022). However, since existing DFIS methods generate images without prior knowledge of natural images, the image search space becomes huge. Consequently, they produce images with unnatural or artificial patterns that deviate from the training data distribution, ultimately limiting the model's utility.

In this paper, we introduce a novel DFIS paradigm that leverages the inherent natural image understanding of an off-the-shelf Text-to-Image (T2I) diffusion model. Recent breakthroughs in T2I diffusion models trained on large-scale open-world datasets (Rombach et al., 2022; Saharia et al., 2022) provide powerful image priors capable of significantly narrowing the search space on DFIS. However, naively replacing the original training data with images generated by the T2I diffusion model is insufficient. Due to the inaccessibility of information regarding the training dataset, encompassing its domain, class-specific attributes, the input prompts of the T2I diffusion model are necessarily impre-

---

[*]Equal contribution  [1]Korea University  [2]Korea Institute of Science and Technology  [3]Korea Advanced Institute of Science and Technology  [4]Kyung Hee University. Correspondence to: Suhyun Kim <dr.suhyun.kim@gmail.com>.

*Proceedings of the 42ⁿᵈ International Conference on Machine Learning*, Vancouver, Canada. PMLR 267, 2025. Copyright 2025 by the author(s).

cise (e.g., A dog). This vagueness leads to the generation of countless images that do not align with the training set distribution. Consequently, without a guiding mechanism based on the pre-trained model's knowledge of its training data distribution, the generated images are highly susceptible to deviating from the underlying distribution, ultimately limiting the model's utility in various applications.

Therefore, we propose a *Diffusion-assisted Data-free Image Synthesis* (DDIS), which guides the T2I diffusion model to generate images that are closely aligned with the training set distribution. Our approach tackles the misalignment problem that arises when directly substituting the training set with images synthesized by a T2I diffusion model. Specifically, our method focuses on achieving alignment in two key aspects: 1) **domain** and 2) **class-specific detail**.

Firstly, to capture the domain of the training data, we introduce a *Domain Alignment Guidance* (DAG) that guides the image latent during the diffusion sampling process. We are inspired by the understanding that Batch Normalization (BN) layers encode the domain knowledge of the training set (Wu et al., 2024; Wang et al., 2020). Specifically, the running statistics in BN layers, derived from the mean and variance of all training batches, capture the distribution of the entire training set, including its domain knowledge. Consequently, DAG guides the diffusion model in aligning the internal statistics of the synthesized samples with the running statistics within the given model. This alignment effectively induces and strengthens the domain-specific features of the training dataset in the synthesized images.

Secondly, we propose to find a new embedding vector corresponding to a pseudo-word, *Class Alignment Token* (CAT), that represents specific concepts of the target class. CAT is a token designed to capture the class details of training data that are not revealed on the class label name itself. For example, while a class label might be 'dog,' the specific dog breeds within the training set are not specified. We aim to find the embedding vector of CAT, ensuring that the synthetic images accurately align with the target class the model was trained on. Interestingly, CAT can not only capture detailed class features but also resolve the lexical ambiguity of class labels. We observe that CAT generates images precisely associated with the intended class when a class label is a homonym or a compound noun.

Conclusively, our proposed method, leveraging a T2I diffusion model as a strong image prior, synthesizes samples closely aligned with the training set distribution. DDIS enables replacing inaccessible original training data, ultimately enhancing the pre-trained model's utility. DDIS outperforms existing DFIS methods in capturing the training data distribution across various domains, such as art, cartoons, and manga, as well as across the various 1000 classes within ImageNet-1k. Moreover, DDIS achieves state-of-the-

art results in data-free knowledge distillation and pruning, verifying the benefit of our synthesized data in enhancing the given model's utility. Our key contributions are:

- We propose a *DDIS*, a novel Data-Free Image Synthesis (DFIS) method that pioneers the use of a Text-to-Image (T2I) diffusion model as a strong image prior to address the issue of existing DFIS methods producing images distant from the training set distribution due to an vast image search space.

- To ensure alignment at both the domain and class levels, we introduce *Domain Alignment Guidance* (DAG) and *Class Alignment Token* (CAT). DAG guides diffusion sampling to align the statistics of generated images with the internal statistics of the pre-trained model and CAT embedding encodes class-specific details.

- Experimental results show that DDIS outperforms prior DFIS methods by generating samples closely aligned with the original training set, leading to state-of-the-art performance in various data-free applications.

## 2. Related Works

### 2.1. Data-Free Image Synthesis (DFIS)

DFIS aims to approximate the distribution $p(x)$ learned by a given classifier by solving an optimization problem in the input space without accessing the training data (Kim et al., 2022). DeepDream (Mordvintsev et al., 2015) visualizes the patterns learned by the model for specific classes by iteratively optimizing noise to maximize the output probability for the desired class. DeepInversion (DI) (Yin et al., 2020) tackles the limited image prior in traditional DFIS methods and introduces a regularizer that ensures the statistics of synthetic images follow the running statistics within the model's Batch Normalization (BN) layers (Ioffe & Szegedy, 2015). Plug-In-Inversion (PII) (Ghiasi et al., 2022) addresses the issue of low diversity in generated images by combining various data augmentations during optimization. Despite these successes, finding an optimal $\hat{x}$ that closely approximates $p(x)$ within a vast search space remains a significant challenge without access to the training data. Consequently, the generated images are distant from the training samples.

### 2.2. Diffusion Models

Diffusion models (Ho et al., 2020; Sohl-Dickstein et al., 2015; Song et al., 2020) have garnered considerable attention due to their exceptional sampling quality. Initiating the process with a random noise sample $\mathbf{x}_T \sim \mathcal{N}(0, I)$, the diffusion model iteratively predicts a denoised sample $\mathbf{x}_t$ at each time step $t$. This is known as the reverse process:

$$\mathbf{x}_{t-1} = \mu_{t-1} + \sigma_t \epsilon, \quad \epsilon \sim N(0, I) \qquad (1)$$

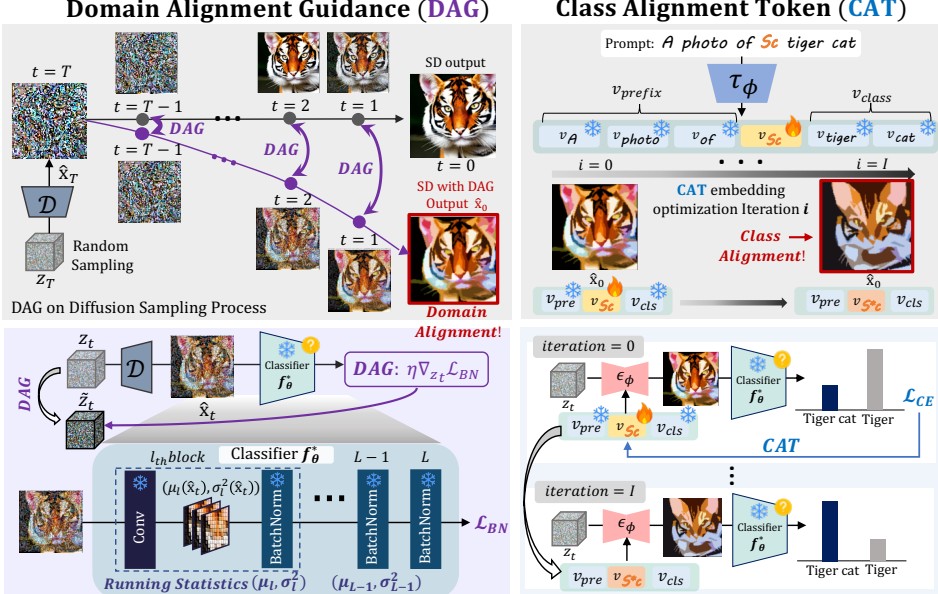

Figure 1. Overall framework of *DDIS*. The goal of DDIS is to generate images approximating the training set distribution learned by $f_\theta^*$ using a Text-to-Image diffusion model. Firstly, we construct prompts $\mathbf{y}$ with *Class Alignment Token* (CAT) and the class label $c$ provided with the model. (e.g., $\mathbf{y} =$ "A/An $\{S_c\}$ $\{class\ label\}$.") Secondly, we provide domain guidance to noise latent $z_t$ at each time step $t$ via *Domain Alignment Guidance* (DAG), aligning image features with the BN layer statistics within a model $f_\theta^*$. Lastly, we forward the final image $\hat{x}_0$ from the guided image latent $\tilde{z}_0$ to the $f_\theta^*$ and optimize the CAT embedding using Cross-Entropy loss to encode features specific to the target class. (As in the figure above, we can successfully synthesize the "tiger cat" class in the "art" domain via DDIS.)

where $\sigma_t$ is noise scale at time $t$ and $\mu_{t-1}$ is given by

$$\mu_{t-1} = \frac{\sqrt{\alpha_{t-1}}\mathbf{x}_t}{\sqrt{\alpha_t}} + (\sqrt{1-\alpha_{t-1}} - \frac{\sqrt{1-\alpha_t}}{\sqrt{\alpha_t}})\epsilon_\theta(\mathbf{x}_t), \quad (2)$$

where $\epsilon_\theta(\mathbf{x}_t)$ is the noise predicted by the U-Net (Ho et al., 2020).

**Sampling Guidance for Diffusion Models.** Classifier-assisted guidance methods contribute to the advancement of diffusion models by allowing the adjustment of outputs to generate desired images. From a score function perspective (Song et al., 2020), we can modify the unconditional score function $\nabla_{\mathbf{x}_t} \log p(\mathbf{x}_t)$ to $\nabla_{\mathbf{x}_t} \log p(\mathbf{x}_t|\mathbf{y})$, controlling image generation to produce results that align with a given condition $\mathbf{y}$, such as a text prompt or class label.

*Classifier Guidance* (CG) (Dhariwal & Nichol, 2021) generates class-conditional images by factorizing $\nabla_{\mathbf{x}_t} \log p(\mathbf{x}_t|\mathbf{y})$ to the unconditional score $\nabla_{\mathbf{x}_t} \log p(\mathbf{x}_t)$ and the gradient derived from the classifier $\nabla_{\mathbf{x}_t} \log p(\mathbf{y}|\mathbf{x}_t)$ by Bayes' rule $p(\mathbf{x}_t|\mathbf{y}) \propto p(\mathbf{y}|\mathbf{x}_t)p(\mathbf{x}_t)$. The classifier guidance can be formulated as the following with a guidance scale $s > 0$:

$$\tilde{\epsilon}_t = \epsilon_\theta(\mathbf{x}_t, \mathbf{y}) - s\sigma_t \nabla_{\mathbf{x}_t} \log p(\mathbf{y}|\mathbf{x}_t). \quad (3)$$

CG requires generating training data $(\mathbf{x}_t, \mathbf{y})$ at each time step $t$ and training a time-dependent external classifier

$p_t(\mathbf{y}|\mathbf{x}_t)$, which is not feasible in DFIS due to the lack of access to labeled datasets.

*Classifier-Free Guidance* (CFG) (Ho & Salimans, 2022) proposes a method for achieving the effects of classifier guidance without the need for an additional classifier by interpolating between unconditional and conditional outputs.

$$\tilde{\epsilon}_t = \epsilon_\theta(\mathbf{x}_t, \mathbf{y}) + s(\epsilon_\theta(\mathbf{x}_t, \mathbf{y}) - \epsilon_\theta(\mathbf{x}_t)) \quad (4)$$

Unfortunately, despite this efficiency, CFG cannot provide guidance on the domain of the training set learned by the classifier, making it insufficient for standalone use in DFIS.

## 3. Method

First, Section 3.1 introduces the preliminaries of data-free image synthesis and diffusion models. Section 3.2 explains the concept of Diffusion-assisted Data-free Image Synthesis (DDIS). Then, Section 3.3 details Domain Alignment Guidance (DAG), which directs the image latent during diffusion sampling, and Class Alignment Token (CAT) in Section 3.4, which captures class details for accurate class alignment.

### 3.1. Preliminary

**Data-Free Image Synthesis.** The goal of Data-Free Image Synthesis (DFIS) is to find the optimal point $\hat{x} \in \mathbb{R}^{B \times C \times W \times H}$ within the high-dimensional input space that

**Algorithm 1** Domain Alignment Guidance Algorithm

**Require:** `Model`(Pre-trained unconditional score estimator $\epsilon_\theta(\cdot, t)$, text encoder $\tau_\phi$, image decoder $\mathcal{D}$, prompt $\mathbf{y}$, classifier $f(\cdot; \theta^*)$ pre-trained on unknown dataset

**Output:** Latent $\hat{\mathbf{z}}_0$

1: $\mathbf{z}_T \sim \mathcal{N}(0, I)$
2: **for** $t = T - 1$ **to** $1$ **do**
3: $\quad \epsilon_t \sim \mathcal{N}(0, I)$
4: $\quad \mathbf{z}_t \leftarrow \mu_t + \sigma_{t+1}\epsilon_{t+1}$ {obtain $\mu_t$ from Equation 2}
5: $\quad \mathcal{L}_{BN}(\mathcal{D}(\mathbf{z}_t)) = \sum_{l=1}^{L}(\|\mu_l(\hat{\mathbf{x}}_t) - \mu_l\|_2 + \|\sigma_l^2(\hat{\mathbf{x}}_t) - \sigma_l^2\|_2)$
6: $\quad \tilde{\mathbf{z}}_t \leftarrow \mathbf{z}_t - \eta\nabla_{\mathbf{z}_t}\mathcal{L}_{BN}(\mathcal{D}(\mathbf{z}_t))$ {Apply DAG}
7: $\quad \tilde{\epsilon}_t \leftarrow \epsilon_\theta(\tilde{\mathbf{z}}_t; \varnothing, t) + s(\epsilon_\theta(\tilde{\mathbf{z}}_t; \tau_\phi(\mathbf{y}), t) - \epsilon_\theta(\tilde{\mathbf{z}}_t; \varnothing, t))$
8: $\quad \tilde{\mathbf{z}}_{t-1} \leftarrow \sqrt{\bar{a}_{t-1}}\left(\frac{\tilde{\mathbf{z}}_t + (1-\bar{a}_t)\tilde{\epsilon}_t}{\sqrt{\bar{a}_t}}\right) + \sqrt{1 - \bar{a}_{t-1} - \sigma_t^2}\tilde{\epsilon}_t + \sigma_t\epsilon_t$
9: **end for**
10: **Return** $\tilde{\mathbf{z}}_0$

---

**Algorithm 2** DDIS Algorithm

**Require:** Pre-trained T2I diffusion model `Model`, classifier $f(\cdot; \theta^*)$ pre-trained on unknown dataset

**Parameter:** Class Alignment token embedding $v_c$

**Output:** Sampling images $\hat{\mathbf{x}}_0$

1: $\mathcal{E}, \mathcal{D}, \tau_\phi, \epsilon_\theta \leftarrow$`Model`
2: **for** number of total classes $C$ **do**
3: $\quad$ Define the prompt $\mathbf{y}$
4: $\quad$ (e.g $\mathbf{y} = $ "`A`/`An`$\{S_c\}\{c_{th}\ class\ label\}$")
5: $\quad$ Add $S_c$ to vocabulary and get the $v_c$ from $\tau_\phi(S_c)$
6: $\quad$ **for** number of iterations $i$ **do**
7: $\quad\quad \tilde{\mathbf{z}}_0 \leftarrow DomainAlignmentGuidance(\text{Model}, \mathbf{y})$
8: $\quad\quad \hat{\mathbf{x}}_0 \leftarrow \mathcal{D}(\tilde{\mathbf{z}}_0)$
9: $\quad\quad v_c \leftarrow \mathcal{L}_{CE}(f(\hat{\mathbf{x}}_0; \theta^*), \mathbf{c})$ {Optimize the $v_c$}
10: $\quad$ **end for**
11: $\quad$ Save the $c_{th}$ optimal Class Alignment token embedding $v_c^*$
12: **end for**
13: **Return** $\hat{\mathbf{x}}_0 \in \{\hat{\mathbf{x}}_0^1, \hat{\mathbf{x}}_0^2 \cdots \hat{\mathbf{x}}_0^C\}$

---

elicits a maximum response from the classifier for a desired class $p(y|\hat{x})$ under data-free conditions. This process involves iteratively updating a random noise into a visually natural image by optimizing the following loss function:

$$\min_{\hat{x}} \mathcal{L}(\hat{x}, y) + \mathcal{R}(\hat{x}), \quad (5)$$

where $\mathcal{L}(\cdot)$ is Cross-Entropy loss, and $\mathcal{R}(\cdot)$ is an image prior regularizer. Despite the potential of DFIS, finding an optimal point in a vast search space requires huge computational costs under data-free settings. Although DeepDream (Mordvintsev et al., 2015) explores a total variation regularizer to generate visually plausible outputs, the resulting images lack the naturalness and fidelity of the real samples.

**Text Conditioned Latent Diffusion Models.** Text Conditioned Latent Diffusion models (LDM) (Rombach et al., 2022) operate in the latent space by projecting the image $\mathbf{x}_0$ to $\mathbf{z}_0 = \mathcal{E}(\mathbf{x}_0)$ through the auto-encoder $\mathcal{E} : \mathbb{R}^k \to \mathbb{R}^d$. In the diffusion sampling process, starting from $\mathbf{z}_T$, we

subsequently obtain the denoised latent and finally generate the clean image $\mathbf{x}_0$ by passing the $\mathbf{z}_0$ to the decoder $\mathcal{D} : \mathbb{R}^d \to \mathbb{R}^k$. Expressly, since the text conditioned LDMs focus on generating images guided by the given text prompt $\mathbf{y}$ encoded by the text encoder $\tau_\phi$ as a condition, and the predicted noise is $\epsilon_\theta(\mathbf{z}_t; \tau_\phi(\mathbf{y}), t)$.

**Guidance for Text-to-Image Diffusion Models.** As mentioned in Section 2.2, Classifier-Free Guidance (CFG), allows the class label or text to be used as a condition in the diffusion process can be represented in the latent space by modifying Equation 4 as follows:

$$\tilde{\epsilon}_t = \epsilon_\theta(\mathbf{z}_t; \varnothing, t) + s(\epsilon_\theta(\mathbf{z}_t; \tau_\phi(\mathbf{y}), t) - \epsilon_\theta(\mathbf{z}_t; \varnothing, t)), \quad (6)$$

where $\varnothing$ means a null condition (unconditioned) and $\mathbf{z}_t$ is latent variable in time step $t$ given by

$$\mathbf{z}_{t-1} = \sqrt{\bar{a}_{t-1}}\left(\frac{\mathbf{z}_t + (1-\bar{a}_t)\epsilon_\phi^t}{\sqrt{\bar{a}_t}}\right) + \sqrt{1 - \bar{a}_{t-1} - \sigma_t^2}\epsilon_\phi^t + \sigma_t\epsilon_t \quad (7)$$

## 3.2. Diffusion-Assisted Data-free Image Synthesis

In this section, we introduce Diffusion-assisted Data-free Image Synthesis (DDIS), which is a pioneer of the use of a Text-to-Image (T2I) diffusion model as a strong image prior to address the issue of existing DFIS methods producing images distant from the training set distribution due to a vast image search space. We utilize the Stable Diffusion (SD) (Rombach et al., 2022), which has achieved remarkable success in T2I diffusion models. By leveraging the comprehensive knowledge of natural images embedded in this pre-trained T2I diffusion model, DDIS can produce samples that better approximate the learned distribution than current DFIS methods. However, since the images generated by T2I diffusion models still differ from the exact training set distribution, additional guidance is necessary to ensure they align properly with the training set.

## 3.3. Domain Alignment Guidance

We propose an effectively applicable guidance method for the diffusion model, called *Domain Alignment Guidance* (DAG), which provides guidance on the domain learned by the classifier during the diffusion sampling process. Inspired by studies suggesting that Batch Normalization (BN) layers encode domain knowledge (Lim et al., 2023; Mirza et al., 2022), Domain Alignment Guidance (DAG) guides the statistics of synthesized images to align with the training set statistics stored in the model's BN layers. Specifically, we can obtain the channel-wise running mean $\mu$ and running variance $\sigma^2$ of the entire training set from all BN layers in the given model $f(\cdot; \theta^*)$. To guide the image latent so that the statistics of the generated images follow the running statistics, we modify the unconditional score of the

image latent $\nabla_{\mathbf{z}_t} \log p(\mathbf{z}_t)$ to $\nabla_{\mathbf{z}_t} \log p(\mathbf{z}_t|\mu, \sigma^2)$ from the perspective of SDE (Song et al., 2020). Using Bayes' rule, the conditional score can be factorized as below two terms:

$$\nabla_{\mathbf{z}_t} \log p(\mathbf{z}_t|\mu, \sigma^2) = \nabla_{\mathbf{z}_t} \log p(\mathbf{z}_t) + s\nabla_{\mathbf{z}_t} \log p(\mu, \sigma^2|\mathbf{z}_t), \quad (8)$$

where $s$ is a parameter controlling the guidance strength. We can interpret the second term as the gradient with respect to the latent obtained from an external loss function, which minimizes the difference between the running statistics and those of the synthetic image:

$$\mathcal{L}_{BN}(\hat{\mathbf{x}}_t) = \sum_{l=1}^{L}(\|\mu_l(\hat{\mathbf{x}}_t) - \mu_l\|_2 + \|\sigma_l^2(\hat{\mathbf{x}}_t) - \sigma_l^2\|_2), \quad (9)$$

where $\mu_l(\hat{\mathbf{x}}_t)$ and $\sigma_l^2(\hat{\mathbf{x}}_t)$ represent the mean and variance of the feature map at the $l$-th layer of the generated image $\hat{\mathbf{x}}_t$. In this process, we project the image latent to the pixel space at each time step $t$, by $\hat{\mathbf{x}}_t = \mathcal{D}(\mathbf{z}_t)$, to compute the gradient. We replace $\nabla_{\mathbf{z}_t} \log p(\mu, \sigma^2|\mathbf{z}_t)$ with the obtained gradient $\nabla_{\mathbf{z}_t} \mathcal{L}_{BN}(\mathcal{D}(\mathbf{z}_t))$ w.r.t the latent variable, thereby providing guidance to the image latent in the direction of the training set distribution as below:

$$\tilde{\mathbf{z}}_t = \mathbf{z}_t - \eta\nabla_{\mathbf{z}_t}\mathcal{L}_{BN}(\mathcal{D}(\mathbf{z}_t)), \quad (10)$$

where $\eta$ is the scaling factor for the gradient influence. Finally, the DAG is integrated with the CFG to guide the latent toward the target distribution, enabling the T2I diffusion model to sample images faithful to the prompt while incorporating domain knowledge within the classifier.

$$\tilde{\epsilon}_t = \epsilon_\theta(\tilde{\mathbf{z}}_t; \varnothing, t) + s(\epsilon_\theta(\tilde{\mathbf{z}}_t; \tau_\phi(\mathbf{y}), t) - \epsilon_\theta(\tilde{\mathbf{z}}_t; \varnothing, t)) \quad (11)$$

In conclusion, starting from noise $\mathbf{z}_T \sim \mathcal{N}(0, I)$, we estimate the noise $\tilde{\epsilon}_t$ based on guided latent $\tilde{\mathbf{z}}_t$ via the DAG. This process iteratively updates the latent to a cleaner one, ultimately sampling images $\hat{\mathbf{x}}_0$ that is aligned with the target domain by modifying Equation 7.

$$\tilde{\mathbf{z}}_{t-1} = \sqrt{\bar{a}_{t-1}}\left(\frac{\tilde{\mathbf{z}}_t + (1-\bar{a}_t)\tilde{\epsilon}_t}{\sqrt{\bar{a}_t}}\right) + \sqrt{1-\bar{a}_{t-1}-\sigma_t^2}\tilde{\epsilon}_t + \sigma_t\epsilon_t \quad (12)$$

The overall algorithm of DAG is described in Algorithm 1.

### 3.4. Class Alignment Token

To capture the accurate class's property, we encode the semantic information about class details into the word embedding of the proposed *Class Alignment Token* (CAT). In a T2I diffusion model, words or sub-words from the input prompt are converted into tokens from a predefined vocabulary, and a pre-trained text encoder $\tau_\phi$ maps each token to a unique word embedding. We chose this embedding space for optimization in DDIS because this space encodes rich semantic representations learned by the pre-trained text

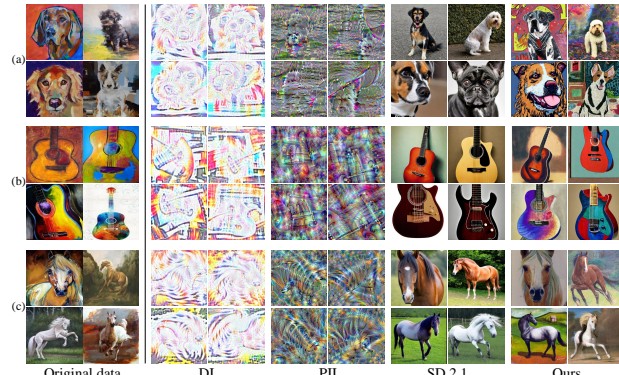

*Figure 2.* Qualitative comparison with various DFIS methods on the PACS *(Art Painting)* dataset. (a)-(c) correspond to the classes dog, guitar and horse, respectively.

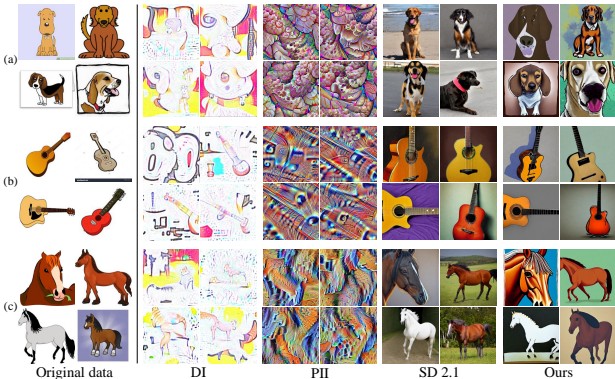

*Figure 3.* Qualitative comparison with various DFIS methods on the PACS *(Cartoon)* dataset. The classes match those in Figure 2.

encoder, which we expect to capture the semantic features of the desired class effectively. Inspired by (Gal et al., 2022; Huang et al., 2024; Kim et al., 2025), which encodes specific concepts into pseudo-word token embeddings, we define $S_c$ as a CAT, to precisely represent the desired class $c$. Then, we construct a simple prompt $\mathbf{y}$ using the given class label as $\mathbf{y} =$"A/An $\{S_c\}$ $\{class\ label\}$". We expand the vocabulary by adding the CAT and only optimize its token embedding $v_c$ of the $\{S_c\}$ using Cross-Entropy (C.E.) loss. This iterative process aims to find the optimal $v_c^*$ that captures the class-specific information learned by the model, as follows:

$$\mathcal{L}_{CE}(f(\hat{\mathbf{x}}_0; \theta^*), \mathbf{c}) = -\sum_{j=1}^{N} c_j \log\left(\sigma(f_j(\hat{\mathbf{x}}_0; \theta^*))\right), \quad (13)$$

where $f(\hat{\mathbf{x}}_0; \theta^*)$ indicates output logits given the pre-trained model $f(\cdot; \theta^*)$ and $\hat{\mathbf{x}}_0 = \mathcal{D}(\mathbf{z}_0)$. Specifically, we compute the C.E. loss only for the final image $\hat{\mathbf{x}}_0$ obtained through the DAG-based diffusion sampling process up to the final time step, as this image closely resembles the distribution observed by the classifier. (i.e. $p(\mathbf{x}) \approx p(\hat{\mathbf{x}}_0)$). Thus,

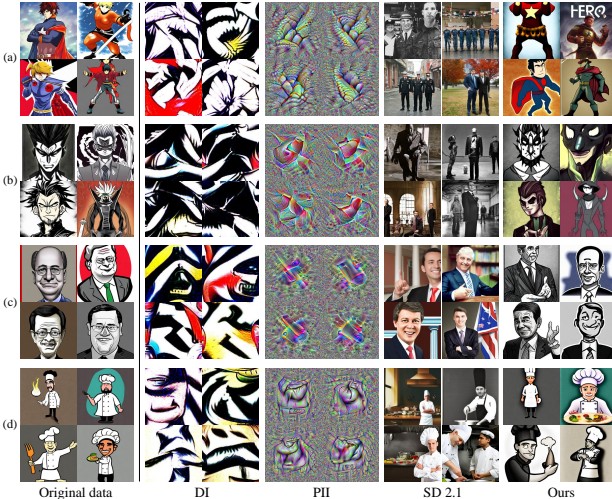

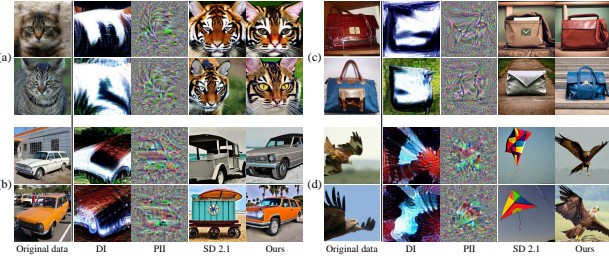

*Figure 5.* Qualitative comparison with prior methods on the ImageNet-1k dataset. We also include comparisons with StableDiffusion 2.1 (SD2.1), which serves as the baseline model in our work. (a-d) denote classes "tiger cat", "beach wagon", "mail bag" and "kite", respectively. It is noteworthy that our method can address the issue of *lexical overlapping*.

*Figure 4.* Qualitative comparison with baseline methods on the StyleAligned dataset. We conduct experiments on two domains, Manga (a,b) and Caricature (c,d). (a-d) correspond to the classes hero, villain, politician, and chef, respectively.

### 4.2. Main Results

**Image Synthesis with Various Domain.** DDIS is the first to succeed in generating samples from various domains, not just the photo domain, in the DFIS. Figures 2 to 4 compare images generated with the models pre-trained on diverse domain datasets with baselines. Since we do not know the knowledge of the dataset used for training the classifier, existing DFIS methods must explore an extremely large image search space. This leads to generated images with artifacts that fail to accurately capture the training dataset's domain properties. As a result, they fail to capture the domain properties of the training dataset and usually generate images with severe artifact effects. Although Stable Diffusion can generate class-faithful images given prompts such as 'A dog', it lacks guidance on the domain of the training data, preventing it from capturing the dataset's domain-specific information. In contrast, the proposed DDIS leverages DAG in the diffusion sampling process and optimizes CAT embeddings to synthesize images that accurately reflect the training dataset's domain and class attributes, producing images that closely resemble the original data.

even without access to the training set or additional training processes, we can capture the class-specific attribute by optimizing only the CAT embedding. Furthermore, once the optimal CAT embedding is found, we can quickly generate images of the desired class through a sampling process. Consequently, DDIS can generate images that approximate the training set via DAG in the sampling process and CAT optimization in the T2I diffusion model, ensuring significant improvements in image quality compared to existing DFIS methods. The overall framework is described in Figure 1, and the algorithm can be found in Algorithm 2.

**Image Synthesis Faithful to the Target Class.** We demonstrate how CAT embedding optimization enhances the capture of target class attributes. Figure 5 presents images generated with a ResNet-34 pre-trained on ImageNet-1k. Surprisingly, DDIS achieves precise class mappings, even resolving the lexical overlap problem, where generators misinterpret class labels. For instance, suppose the classifier has learned the tiger cat, as shown in Figure 5 (a) (though we do not know the training set). However, Stable Diffusion (SD) incorrectly generates a tiger due to word similarity. This issue becomes even more pronounced with compound nouns, such as 'beach wagon' or 'mailbag' where ambiguity leads to incorrect images. Even for single-word labels (Figure 5 (d)), SD struggles with homonyms, reducing accuracy. In contrast, DDIS overcomes these challenges

## 4. Experiments

### 4.1. Experimental Settings

**Baselines.** We mainly compare our method with existing data-free image synthesis (DFIS) methods such as DeepInversion (DI) (Yin et al., 2020) and PlugInInversion (PII) (Ghiasi et al., 2022), which deal with the large-scale datasets synthesis inverting the CNNs. We do not compare the performance the NaturalInversion (NI) (Kim et al., 2022) since NI only considers small-scale datasets.

**Datasets.** We utilize ResNets (He et al., 2016) trained on five domains to generate images. For the photo domain, we use ImageNet-1k (Russakovsky et al., 2015). For the art painting and cartoon domain, we use the PACS dataset (Li et al., 2017). Additionally, we collect artificial datasets via Style Aligned (Hertz et al., 2024) for manga and caricature domains. Each dataset includes 7 classes. Detailed experimental settings and class labels are in the appendix.

*Table 1.* Comparison of generated image quality with baselines. We evaluate how closely generated images approximate the training set distribution and further measure the fidelity and diversity of images using Precision and Recall metrics.

| Dataset | Domain | DDIS (Ours) | | | | DeepInversion | | | | PlugInInversion | | | |
|---|---|---|---|---|---|---|---|---|---|---|---|---|---|
| | | IS(↑) | FID(↓) | Precision(↑) | Recall(↑) | IS(↑) | FID(↓) | Precision(↑) | Recall(↑) | IS(↑) | FID(↓) | Precision(↑) | Recall(↑) |
| ImageNet-1k | Photo | 15.92 | 30.31 | 0.8028 | 0.7731 | 9.52 | 187.63 | 0.5038 | 0.0907 | 3.51 | 220.62 | 0.2969 | 0.0528 |
| PACS | Art painting | 4.12 | 133.37 | 0.7742 | 0.3213 | 4.00 | 188.53 | 0.4033 | 0.0004 | 2.53 | 208.73 | 0.3442 | 0.0014 |
| | Cartoon | 4.04 | 85.41 | 0.7541 | 0.3104 | 3.91 | 148.94 | 0.3704 | 0.0038 | 2.81 | 275.86 | 0.0004 | 0.0001 |
| Style-Aligned | Caricature | 3.94 | 139.75 | 0.3552 | 0.5672 | 3.58 | 195.25 | 0.1027 | 0.0618 | 2.51 | 293.58 | 0.0001 | 0.0004 |
| | Manga | 3.87 | 145.82 | 0.2036 | 0.4396 | 3.32 | 206.57 | 0.0834 | 0.0019 | 2.36 | 295.14 | 0.0001 | 0.0003 |

*Table 2.* Data-Free Knowledge Distillation (DFKD) performance on PACS and ImageNet-1k with synthesized images from various DFIS methods. 'T.' and 'S.' denote the teacher and student networks, respectively. 'Original' refers to the baseline accuracy of a student network trained on the original training set.

| Dataset | T. | S. | Original | DI | PII | SD | Ours |
|---|---|---|---|---|---|---|---|
| PACS-art | ResNet-34 | ResNet-18 | 32.60 | 17.49 | 12.27 | 21.93 | **28.46** |
| | ResNet-34 | VGG-11 | 26.03 | 12.65 | 9.23 | 18.12 | **21.89** |
| | ResNet-50 | ResNet-18 | 37.98 | 18.48 | 15.32 | 31.67 | **35.87** |
| | ResNet-50 | ResNet-34 | 28.46 | 16.63 | 13.36 | 19.52 | **24.65** |
| | VGG-16 | VGG-11 | 32.89 | 17.37 | 12.47 | 22.61 | **27.43** |
| | VGG-16 | ShuffleNetv2 | 54.25 | 30.32 | 23.14 | 41.74 | **48.56** |
| PACS-cartoon | ResNet-34 | ResNet-18 | 51.35 | 28.65 | 17.23 | 38.12 | **47.06** |
| | ResNet-34 | VGG-11 | 49.95 | 26.92 | 15.62 | 38.94 | **44.92** |
| | ResNet-50 | ResNet-18 | 58.15 | 31.48 | 20.32 | 44.67 | **51.45** |
| | ResNet-50 | ResNet-34 | 47.34 | 24.05 | 15.66 | 36.73 | **42.93** |
| | VGG-16 | VGG-11 | 48.82 | 25.29 | 16.04 | 38.15 | **44.81** |
| | VGG-16 | ShuffleNetv2 | 54.24 | 29.94 | 19.32 | 43.63 | **49.32** |
| ImageNet-1k | ResNet-34 | ResNet-18 | 43.30 | 4.67 | 2.01 | 33.02 | **41.68** |
| | ResNet-34 | VGG-11 | 34.91 | 2.67 | 1.34 | 27.41 | **32.71** |
| | ResNet-50 | ResNet-18 | 43.09 | 6.31 | 1.98 | 35.43 | **40.77** |
| | ResNet-50 | ResNet-34 | 44.57 | 7.29 | 3.08 | 33.15 | **42.87** |
| | VGG-16 | VGG-11 | 34.70 | 2.55 | 1.23 | 26.93 | **32.14** |
| | VGG-16 | ShuffleNetv2 | 28.13 | 1.37 | 1.02 | 18.52 | **24.61** |

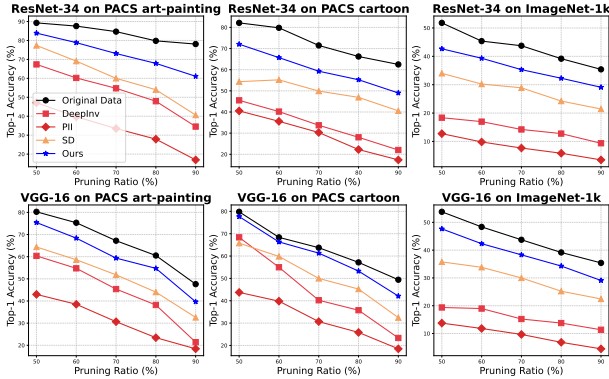

*Figure 6.* Finetuning the pruned ResNet-34 and VGG-16 with synthetic PACS and ImageNet-1k samples under different pruning ratios. We compare DDIS performance with prior DFIS methods.

by optimizing only the CAT embedding, ensuring that the generated images accurately align with the intended class.

**Quantitative Results.** We compare the image quality with existing DFIS studies to evaluate whether the generated images approximate the distribution of the training set used to train the model. For this evaluation, we use (1) Incep-

tion Score (IS) (Reed et al., 2016) and Frechet Inception Distance (FID) (Heusel et al., 2017), and (2) Precision and Recall (P&R) (Sajjadi et al., 2018), to measure the fidelity and diversity of synthetic images strictly. We evaluate the image quality of 10,000 synthetic ImageNet-1k samples, 2,800 synthetic PACS samples, and 1,400 synthetic Style-Aligned samples. As shown in Table 1, DDIS outperforms the baselines across all metrics. Our proposed method generates images that most closely approximate the training set distribution, effectively producing images that align with both the class and domain of the training set.

### 4.3. Data-Free Applications

DDIS aims to enhance the utility of a given model by generating samples that approximate the distribution of the training data. Accordingly, we conduct experiments on Knowledge Distillation (KD) and Pruning using synthetic images without direct access to the training data. Specifically, we synthesize 2,800 images from the PACS dataset and 100k images from ImageNet-1k for these experiments.

**Data-Free Knowledge Distillation.** In this section, we ensure that we can transfer information from a teacher network to a student network without original data and outperform existing data-free knowledge distillation approaches. The experimental setup for knowledge distillation is based on the protocol outlined in (Li et al., 2023). Our method is compared against previous DFIS approaches, including DI and PII, as well as the use of images generated by Stable Diffusion (SD).

2 demonstrates the superior performance of the proposed approach. Our method consistently achieves results closer to the baseline across all datasets compared to prior studies. Notably, it better approximates the training set distribution than directly using SD outputs as training data. This suggests that our method generates samples that are more aligned with the domain and class distributions of the original training data.

**Data-Free Pruning.** We demonstrate that DDIS enhances pruned model accuracy without using real data. We apply L1-norm pruning (Liu et al., 2017) to ResNet-34 and VGG-

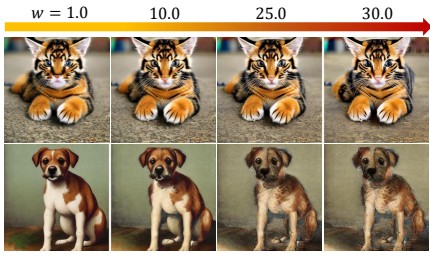

*Figure 7.* Sample visualization of the re-weighting of the optimal Class Alignment Token embedding. We visualize synthetic samples shown as the scaling parameter $w$ of the CAT embedding vector progressively increases from 1.0 to 30.0.

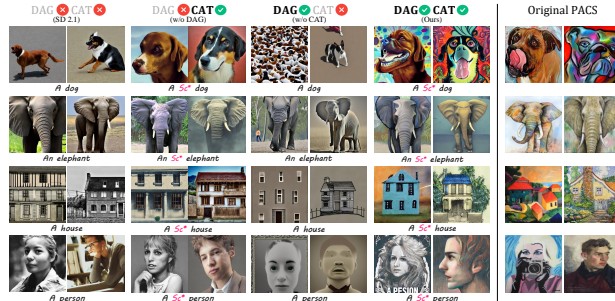

*Figure 8.* Ablation results on PACS *(Art painting)* dataset. We assess the impact of the proposed DAG and CAT on synthetic images. Note that the case where neither DAG nor CAT is used is equivalent to using the standard SD 2.1.

16 on PACS art painting, cartoon and ImageNet-1k with pruning ratios from 50% to 90%. Following (Liu et al., 2018) settings, we locally prune the least important channels in each layer at the specified ratios. Similar to DFKD results, DDIS synthesizes samples closely matching the underlying distribution, enhancing model utility, as shown in Figure 6.

### 4.4. Further Analysis

**Ablation Study.** To understand the impact of each component of DDIS on image synthesis, we evaluate the performance of various component combinations on the PACS art painting dataset, as described in Figure 8 and Table 3. When directly generating images from vanilla SD, the resulting images lack the information of the training set and fail to capture the unique characteristics of each class and domain, leading to significant deviations from the training set. The DAG leads to synthetic images that better reflect the target domain by following the inner statistics of the training set encapsulated in the model. The optimal embedding of the CAT captures class-specific features, generating images with high fidelity to the class label, but if not used with DAG, domain discrepancy issues arise. When all proposed components are used together, the resulting images align with both the domain and class of the training set.

**Impact of Optimal CAT Embedding on Synthetic Image.** We evaluate if the optimal Class Alignment Token (CAT) embedding captures the desired class's high-level semantics and fine details. Figure 7 shows the influence of the CAT embedding vector by scaling the cross-attention map with parameter $w$ from 1.0 to 30.0. As $w$ increases, the CAT embedding accentuates class-specific features, such as the 'eye shape' in the tiger cat images, showing its ability to capture detailed class attributes. Furthermore, strengthening the CAT embedding improves alignment with the target domain, showcasing its capability to represent precise visual details and maintain consistency with the training set distribution.

*Table 3.* Evaluation of image quality in PACS (*Art Painting*) under different elements in our method. We assess the impact of each component on the quality of generated images.

| Method | DAG | CAT | IS(↑) | FID(↓) | Precision(↑) | Recall(↑) |
|---|---|---|---|---|---|---|
| SD | | | 2.88 | 193.57 | 0.6429 | 0.2572 |
| SD w/o DAG | ✓ | | 3.29 | 174.31 | 0.6995 | 0.3074 |
| SD w/o CAT | | ✓ | 3.95 | 166.22 | 0.6871 | 0.2843 |
| **Ours** | ✓ | ✓ | **4.12** | **133.37** | **0.7742** | **0.3213** |

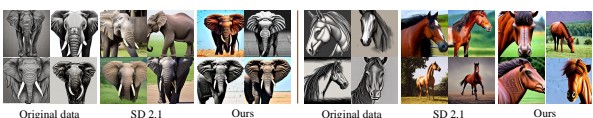

*Figure 9.* Synthesizing the sketch domain is challenging under data-free conditions due to its abstract depiction.

## 5. Limitations

While DDIS excelled across domains, it struggled in the Sketch domain, which has abstract representations of objects and scenes that make image generation particularly challenging, as shown in Figure 9. However, DDIS captures common Sketch characteristics, such as monochrome backgrounds and darker tones. Additionally, since our DAG relies on feature statistics from the BN layer, it can only be applied to models with BN layers. In future work, we plan to explore model-agnostic domain alignment guidance.

## 6. Conclusion

We introduce DDIS, the first Diffusion-assisted Data-free Image Synthesis method, using a T2I diffusion model as a powerful image prior to narrow the image search space. We introduce Domain Alignment Guidance (DAG) for domain alignment during diffusion sampling and Class Alignment Token (CAT) embedding optimization for desired class alignment. We are also the first to conduct experiments across various domains in DFIS, proving its effectiveness.

## Impact Statement

This work aims to enhance the utility of pre-trained models by generating synthetic data in scenarios where access to the original data is unavailable. We acknowledge that data-free image synthesis approaches, designed to recover data following the training dataset's distribution, can raise concerns regarding privacy leakage and other ethical issues. However, our objective is not to reconstruct individual instances or facilitate unauthorized data access but rather to synthesize a surrogate dataset that can be effectively utilized for data-free applications such as data-free knowledge distillation or data-free pruning. Specifically, our method guides the data generation process by leveraging the running statistics from classifiers, which represent the averaged information of the entire dataset. This design inherently prevents the inclusion of individual instance details in the generated images, making it extremely difficult to recover any specific sample (see Figures 2 to 5 in the main paper). We hope that this work contributes to the responsible development of data-free techniques for scenarios in which data sharing is limited or infeasible.

## Acknowledgements

This research was partly supported by the MSIT(Ministry of Science and ICT), Korea, under the ITRC(Information Technology Research Center) support program(IITP-2024-RS-2023-00258649, 50%) supervised by the IITP(Institute for Information & Communications Technology Planning & Evaluation), partly supported by the National Research Foundation of Korea(NRF) grant funded by the Korea government(MSIT) (RS-2025-00562437, 40%), and partly supported by Institute of Information & communications Technology Planning & Evaluation (IITP) grant funded by the Korea government(MSIT) (No.RS-2022-00155911, Artificial Intelligence Convergence Innovation Human Resources Development (Kyung Hee University), 10%).

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

# A. Experiment Settings

### A.1. Hyper parameters

We generate a 512×512 resolution image using Stable Diffusion 2.1 (Rombach et al., 2022) with $T = 30$ diffusion time steps. For applying Domain Alignment Guidance (DAG), we define the gradient scaling factor $\eta$ as the product of $\lambda_{BN}$ and the guidance scale $s_g$. Here, $\lambda_{BN}$ is the scaling factor for $\mathcal{L}_{BN}$, and $s_g$ denotes the DAG guidance scale. These two parameters play a crucial role in our image generation process, and their impact on synthetic images is discussed in the following subsection. DAG is incorporated with the Classifier-Free Guidance (CFG) with a scale of 15.

To define the Class Alignment Token (CAT), we newly define a single token $S_c$ with no inherent meaning and add it to the vocabulary, using its embedding as the initial value. We utilize the token "newcls" as the CAT, but any arbitrary token without meaning can serve as the initial token. To optimize the embedding vector $v_c$ of the CAT, we employ the Adam optimizer with a learning rate of 0.005. We train for up to 30 epochs, each involving 20 gradient accumulation steps and generating images from latent noise initialized with different seeds. It is equivalent to a batch size of 20 and can be adjusted based on resource availability.

### A.2. Detail of $\eta$ in DAG

The main hyperparameter of our proposed Domain Alignment Guidance (DAG) in the diffusion sampling process is $\eta$, which is divided into the gradient flow scale $\lambda_{BN}$ and the DAG guidance scale $s_g$ and we can rewrite the Equation 10 as below.

$$\tilde{\mathbf{z}}_t = \mathbf{z}_t - s_g(\lambda_{BN}(\nabla_{\mathbf{z}_t}\mathcal{L}_{BN}(\mathcal{D}(\mathbf{z}_t)))) \tag{14}$$

To determine the optimal values for the gradient flow scale($\lambda_{BN}$) and guidance scale($s_g$), we conduct experiments across a wide range of values. Specifically, we test $\lambda_{BN}$ between 0.0001 and 1, while the $s_g$ is explored within the range of 0.1 to 100. As a result, we find the optimal settings as a $\lambda_{BN}$ of 0.01 and a $s_g$ of 20. The experimental results are shown in the Figure 11. Users can adjust these two parameters as needed, with the selection of hyperparameters guided by the generated images' confidence score.

### A.3. Training Strategy

**Prompt Design.** Due to the nature of Data-Free Image Synthesis (DFIS), where we have no information about the training set beyond class label information, we design prompts **y** to be very simple and ambiguous (e.g., **y** ="A/An $\{S_c\}$ {*class label*}.") To demonstrate that we can effectively approximate the desired distribution with minimal information, we refrain from using any prefixes like "A photo of ∼."

**Gradient Skipping for CAT embedding optimization.** Gradient backpropagation through all diffusion steps $t$ demands substantial memory. In our experiments, limiting gradient propagation to just the final denoising step (i.e., step $t = 30$ (=final $T$)) reduces the resource usage. Additionally, since the given model was trained on natural images with a similar distribution of images from the final denoising step (i.e., $p(x) \approx p(\hat{x}_0)$), the gradient skipping technique produces an appropriate loss for representing the intended class. Although deeper backpropagation could lead to further improvements, we do not explore this approach due to memory constraints.

**Early Stopping Strategy.** We use a threshold of 0.7 for the proportion of correctly predicted samples in a batch to determine early stopping. During each epoch, the generated batch is evaluated by the classifier. If over 70% of the samples are accurately predicted as the target class, we apply early stopping for the CAT embedding optimization.

### A.4. Computation Overhead

In our experiments using a single RTX 4090 GPU, the optimal CAT embedding is found in approximately 7.5 minutes. Once determined, this embedding remains fixed, enabling rapid image generation via a diffusion sampling process. Compared to existing Data-Free Image Synthesis (DFIS) methods, our DDIS enhances efficiency and cost-effectiveness for large-scale ImageNet-1k dataset generation while preserving high image quality. To illustrate this, we compare the total optimization iterations needed to synthesize 100,000 ImageNet-1k images on a single RTX 4090 GPU, shown in 4.

DeepInversion (DI) (Yin et al., 2020) requires iterative optimization for each mini-batch. With an optimal batch size of 250,

*Table 4.* Comparison of time cost and iteration count for generating 100,000 ImageNet-1k images with prior DFIS methods (DeepInversion, PlugInInversion), and our method.

| Cost | DeepInversion | PlugInInversion | **DDIS (Ours)** |
|---|---|---|---|
| Total Iteration for 100k samples synthesis | 8,000K | 1,120K | 30K |
| Times per 1 iteration (sec) | 0.83 | 0.79 | 15.17 |
| Total training cost (hours) | 18444 | 245 | 126 |

around 20,000 iterations per batch are needed. Generating 100,000 images necessitates repeating this 400 times, totaling 8,000,000 iterations (20,000 iterations per a mini-batch × 400 batches). PlugInInversion (PII) (Ghiasi et al., 2022) optimizes a random input across seven progressive upsampling stages (7×7 to 224×224), with 400 iterations per stage. Generating 100,000 images with a batch size of 250 results in approximately 1,120,000 total iterations (400 iterations per stage × 7 stages × 400 batches).

Our DDIS optimizes only class-wise CAT embedding vectors in a low-dimensional space (1×784). For ImageNet-1k (1000 classes), we perform just 30 iterations per class, totaling 30,000 iterations (30 iterations/class×1000 classes). After finding the CAT embedding, generating 100,000 images involves simply sampling latent vectors without further training or optimization. While CAT optimization has a slightly longer per-iteration time than prior methods, the drastically reduced total iterations make the overall process more efficient.

## B. Sample Visualization

### B.1. ImageNet-1k Sample Synthesis

Since ImageNet-1k (Russakovsky et al., 2015) consists of 1,000 classes, to evaluate whether our DDIS can avoid lexical overlapping issues and accurately align with the correct class, we generate images by inverting the ResNet-34 (Sohl-Dickstein et al., 2015) pretrained on ImageNet-1k provided by TorchVision (Paszke et al., 2019), which has a top-1 accuracy of 73.31%. We generate 10,000 images with 512×512 using the Stable Diffusion 2.1 and resize them to 224×224 for evaluation, matching the typical resolution used during ImageNet-1k training. For comparison, we generate 10,000 images using DeepInversion (DI) (Yin et al., 2020) and PlugInInversion (PII) (Ghiasi et al., 2022), the only Data-Free Image Synthesis (DFIS) studies addressing large-resolution image synthesis, following their official GitHub implementations and proposed their ImageNet-1k parameters. Figure 12 illustrates samples for 30 classes, including those with lexical overlapping issues. The proposed method effectively generates images that precisely match the classes learned by the model through CAT embedding optimization. For example, the class "Yellow Lady's Slipper" (class 986), which refers to a type of flower, demonstrates how DDIS alleviates lexical overlapping problems to produce accurate class images.

### B.2. PACS Sample Synthesis

We are the first to address the synthesis of non-photo-specific domain datasets in DFIS. We focus on the PACS dataset synthesis (Li et al., 2017), a benchmark commonly used in domain generalization, which consists of four domains: Photo, Art Painting, Cartoon, and Sketch.We specifically handle the *Art Painting* and *Cartoon* domains, as the Photo domain's excellence is well-described with the ImageNet-1k experiment, and the Sketch domain represents a failure case in our study. Each domain contains seven classes: dog, elephant, giraffe, guitar, horse, house, and person. We fine-tune the ResNet-34 pre-trained on ImageNet-1k with two domains, excluding the training domain data for inference on the remaining three domains to measure top-1 accuracy. The top-1 accuracies for Art Painting and Cartoon pre-trained ResNet-34 models are 54.73% and 61.78%, respectively. We generate 512×512 images using Stable Diffusion 2.1 and resize them to 224×224 for evaluation, aligning with the typical PACS training resolution. For comparison, we generate 3,000 images using DeepInversion (DI), PlugInInversion (PII), and our DDIS method. We reproduce the DI and PII utilizing the official GitHub implementations and ImageNet-1k parameters matching the PACS dataset resolution.

Figure 13 shows the results of synthesizing all classes in the PACS *Art Painting* domain, while Figure 14 displays the results for the PACS *Cartoon* domain. We demonstrate that our proposed DAG-based diffusion sampling process and CAT embedding optimization align the target dataset's domain and class information more effectively than the baseline methods.

### B.3. Style-Aligned Sample Synthesis

To demonstrate the effectiveness of our methodology across a broader range of domains, we utilize Style-Aligned (Hertz et al., 2024) to create high-quality, domain-specific datasets. We select the domains of manga, caricature, and sketch and synthesized 400 images for each of the seven classes within these domains using Stable Diffusion 2.1. The synthesized datasets are illustrated in the Figure 15. Subsequently, we finetune a classifier using a ResNet-34 pre-trained on ImageNet-1k, following the same approach as with ImageNet-1k and PACS experiments. The top-1 accuracies for manga, caricature, and sketch are 85.70%, 76.80%, and 98.90%, respectively.

## C. Additional Experiments

### C.1. Efficiency of Domain Alignment Guidance

We demonstrate that in DAG, the gradient of $\mathcal{L}_{BN}$, which aligns synthetic image statistics with the BN layer running statistics, provides stable guidance unaffected by the time step. To avoid confusion, we compare our method with Classifier-Guidance (CG) (Dhariwal & Nichol, 2021), which only provides class guidance at the last time step $T$. First, running statistics within all BN layers are derived from the entire training set, making them more robust and stable than providing guidance based on conditional probability gradients for individual samples. Second, we demonstrate that the BN statistics of generated images remain stable across different time steps. Figure 18 illustrates the layer-wise mean and variance of images generated at every time step. We sample 30 images from the "hero" class in the Style Aligned Manga domain over time steps, passing them through a ResNet-34 pre-trained on the Manga dataset to observe the variation in layer-wise mean over time steps. The layer-wise statistics of noise samples at each time step surprisingly exhibit similar trends from time step 0 to $T$, remaining consistent across different time steps. This result indicates that the model's internal statistics remain stable regardless of the time step, demonstrating that guidance aligning synthetic image statistics with BN layer running statistics at each time step is highly reliable.

### C.2. Comparative Analysis of Class/Domain-Wise Token

To demonstrate the effectiveness of our Class Alignment Token, we compare our CAT with the domain-wise token. The domain-wise token is trained to observe the overall classes, while class-wise tokens are trained to observe each class independently. Figure 10 illustrates the results for the PACS-cartoon domain when using a single token embedding for all classes. We observe that the domain-wise token embedding captures the domain knowledge of the training set but generates images with ignored class-specific features, averaging across all classes. In contrast, our CAT embedding encodes class-specific information, capturing precise features corresponding to each class.

### C.3. Zero-shot Image Synthesis

We conduct a zero-shot image synthesis experiment to evaluate whether Stable Diffusion can effectively generate images for unseen classes. First, using Stable Diffusion V3, we select 10 classes that the Stable Diffusion 2.1 model used in our study has never encountered and generate images for those classes with SD V3. Then, we train a ResNet-50 model using the dataset of these 10 classes (see Figure 19). Finally, we utilize Stable Diffusion 2.1 to synthesize the images used for training ResNet-50 with the our proposed DDIS. Surprisingly, as shown in Figure 20, although Stable Diffusion 2.1 struggles with these classes, the CAT embedding can capture the class attributes from the training set and generate images with a distribution similar to that of the original dataset.

### C.4. Further Analysis of CAT Embedding Optimization with Frozen SD Networks

To evaluate the effectiveness of our CAT embedding optimization, we conduct ablation studies on three design choices, synthesizing four lexically ambiguous ImageNet-1K classes: Kite (21), Tiger Cat (282), Beach Wagon (436), and Mail Bag (636) like 5 in the main paper. We compare confidence scores and visual quality under these settings. Firstly, we test three fine-tuning configurations for Stable Diffusion (SD): (b) UNet only, (c) text encoder only, and (d) Full fine-tuning, using the Cross-Entropy (CE) loss used for CAT optimization. As shown in Figure 21 and Table 5, fine-tuned SD produces distorted, low-confidence images, failing to generate class-aligned images. It suggests that the SD fine-tuning in Data-Free Image Synthesis (DFIS) disrupts the prior knowledge within the SD, degrading image quality. Generally, SD is fine-tuned using real data to adapt to specific domains or styles, but in DFIS, the lack of real images leads to unstable training (right side of Figure 21). Moreover, per-class SD fine-tuning leads to individual SD networks, increasing computational cost. In contrast,

*Table 5.* Design Choice 1: Partial or Full Fine-tuning of SD

| Method | C-21 | C-282 | C-436 | C-636 | Avg. Confidence |
|---|---|---|---|---|---|
| (a) **Ours** | 94.12 | 65.94 | 85.73 | 67.66 | 78.36 |
| (b) SD UNet | 0.77 | 0.67 | 2.47 | 1.53 | 1.36 |
| (c) SD Text-Encoder | 0.03 | 21.25 | 17.35 | 2.57 | 10.30 |
| (d) SD Full | 0.26 | 0.40 | 0.01 | 0.01 | 0.17 |

*Table 6.* Design Choice 2: Optimizing Multiple Tokens Embeddings - Confidence Scores

| Method | C-21 | C-282 | C-436 | C-636 | Avg. Confidence |
|---|---|---|---|---|---|
| **1 Token (Ours)** | 94.12 | 65.94 | 85.73 | 67.66 | 78.36 |
| 2 Tokens | 0.01 | 30.19 | 5.04 | 4.10 | 9.83 |
| 3 Tokens | 0.01 | 30.15 | 5.08 | 3.19 | 9.60 |
| 4 Tokens | 0.01 | 30.17 | 5.07 | 3.59 | 9.71 |
| 5 Tokens | 0.01 | 29.14 | 1.32 | 3.54 | 8.50 |

*Table 7.* Design Choice 3: Adding BatchNorm loss on CAT embedding optimization - Confidence Scores

| Method | C-21 | C-282 | C-436 | C-636 | Avg. Confidence |
|---|---|---|---|---|---|
| **Ours** | 94.12 | 65.94 | 85.73 | 67.66 | 78.36 |
| CAT optim. w/BN Loss | 0.01 | 26.15 | 8.26 | 0.03 | 8.61 |

our method freezes SD and optimizes a single token, preserving SD's image prior while efficiently generating high-quality, class-aligned images.

Secondly, we explore whether multiple CAT embeddings improve class expressivity by optimizing embeddings for one to five tokens. As shown in Figure 22 and Table 6, performance drops with more tokens. Lexical ambiguity persists, and confidence scores drop. In data-free settings with simple prompts (e.g., `"A {class}"`), more tokens amplify the effect of randomly initialized embeddings, hindering desired class-aligned image generation. Therefore, a single token is sufficient and more effective for encoding class information in DFIS.

Lastly, we test optimizing CAT embedding with BatchNorm (BN) loss alongside vanilla CE loss. As shown in Figure 23 and Table 7, BN loss negatively affects capturing class semantics by inducing synthetic images toward the averaged statistics of the entire dataset, which causes class mixing and reduces separability. Therefore, optimizing CAT embedding with CE loss alone effectively captures class-specific attributes.

In conclusion, the above design studies validate that desired concepts or complex information can be encoded by optimizing only token embeddings on frozen SD. Building on this, we adopt the idea to DFIS and demonstrate that a single CAT token effectively captures class semantics while preserving SD's priors. Across all design choices, our approach consistently outperforms alternatives, highlighting its effectiveness for DFIS.

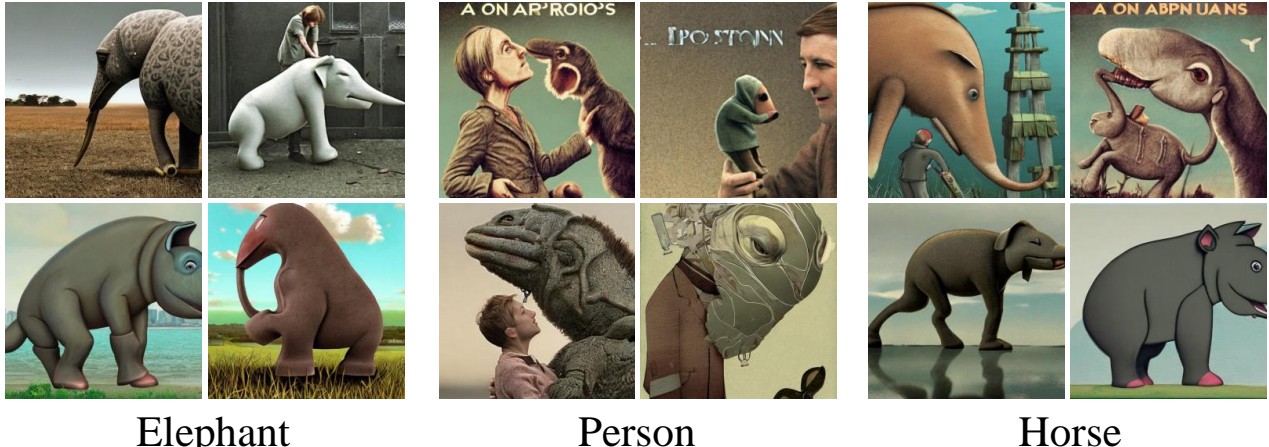

Elephant                    Person                    Horse

*Figure 10.* Visualization of PACS (*cartoon*) samples generated with domain-wise tokens. While domain-wise tokens capture domain information, they fail to distinguish unique class attributes, resulting in averaged class images.

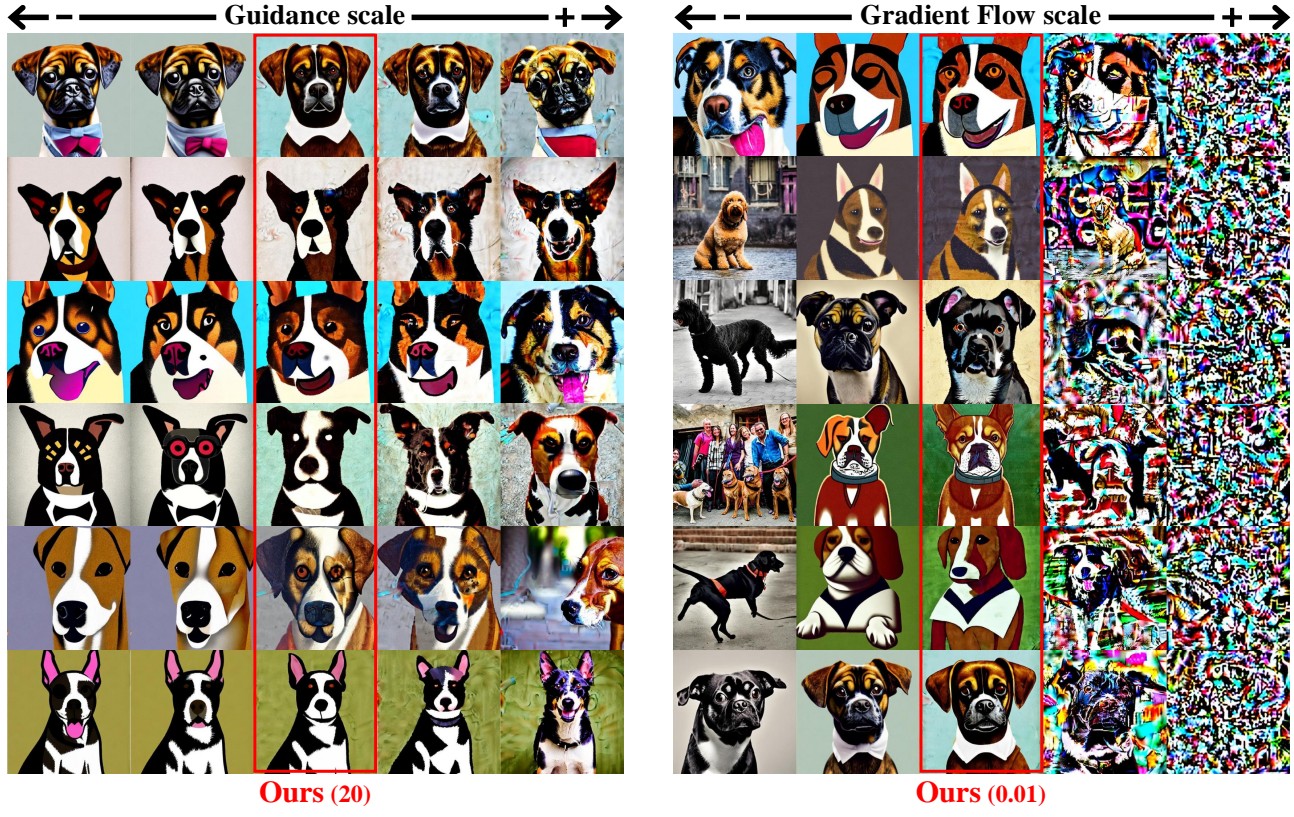

*Figure 11.* Study on Gradient Flow scale and Guidance scale

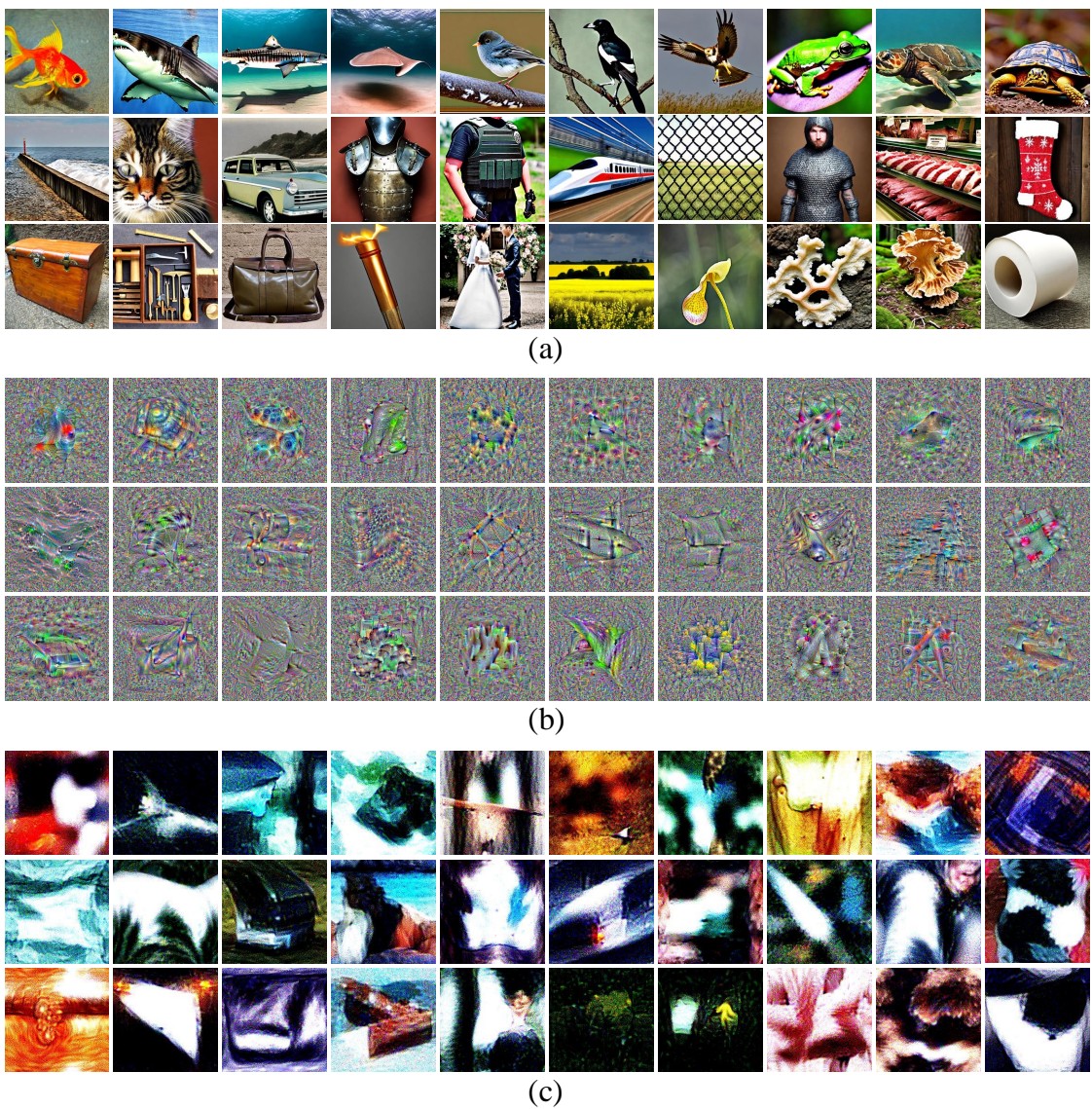

*Figure 12.* Visualization of ImageNet-1k samples. We select 30 ImageNet-1k classes with lexical overlapping issues and present visual results. Panel (a) displays our method, (b) shows PlugInInversion, and (c) depicts DeepInversion. (class index: 1, 2, 3, 6, 13, 18, 21, 31, 33, 37, 282, 436, 460, 461, 465, 466, 467, 489, 490, 492, 496 ,636, 726, 862, 982, 984, 986, 991, 996, and 999)

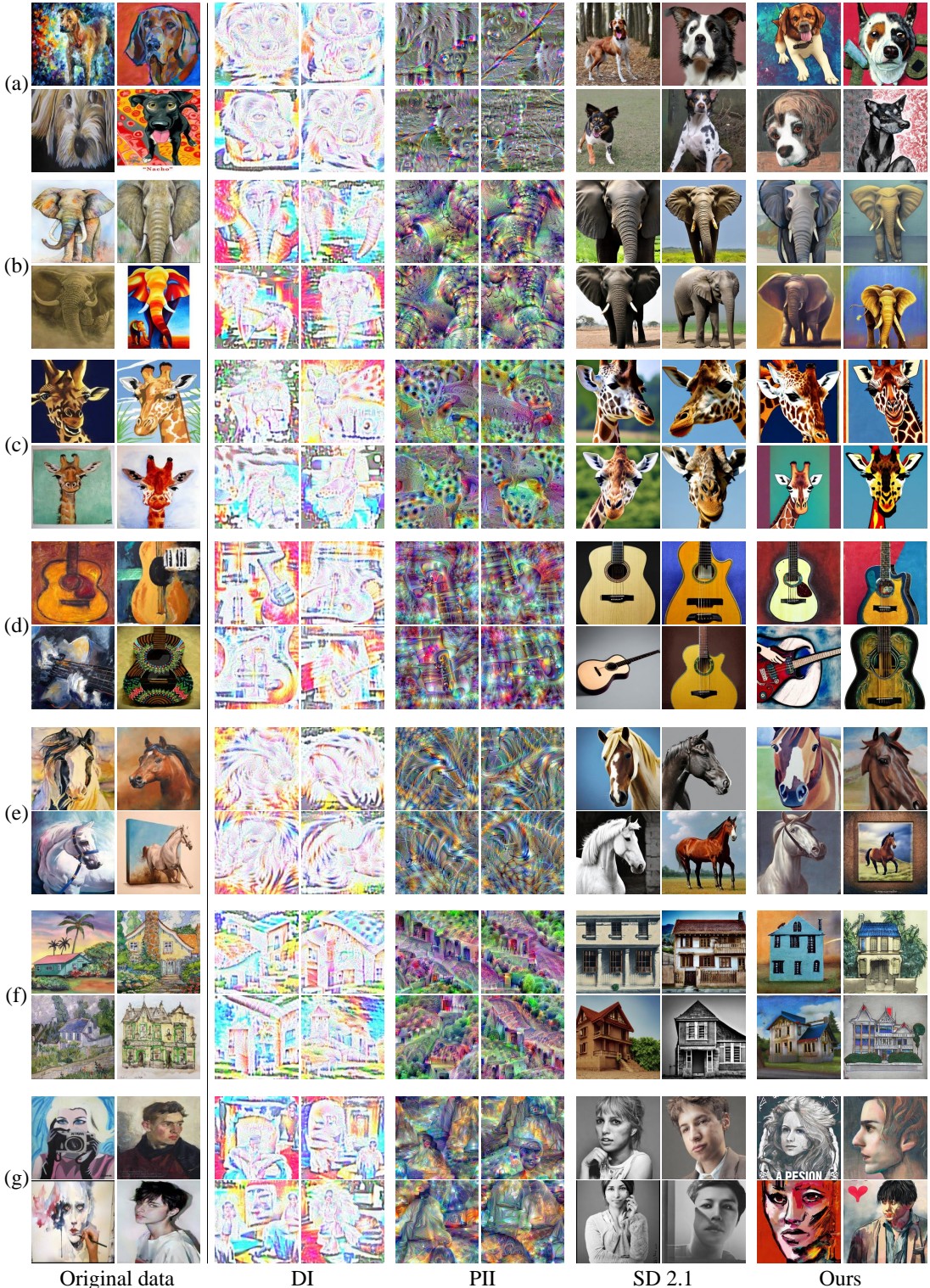

Original data      DI      PII      SD 2.1      Ours

*Figure 13.* Visualization of all class samples in the PACS (*Art Painting*) dataset. (a) through (g) sequentially represent the classes: dog, elephant, giraffe, guitar, horse, house, and person. Our method demonstrates superior capture of target domain and class information compared to baselines (DI, PII) and highlights the domain discrepancy issues that arise when directly applying Stable Diffusion to DFIS.

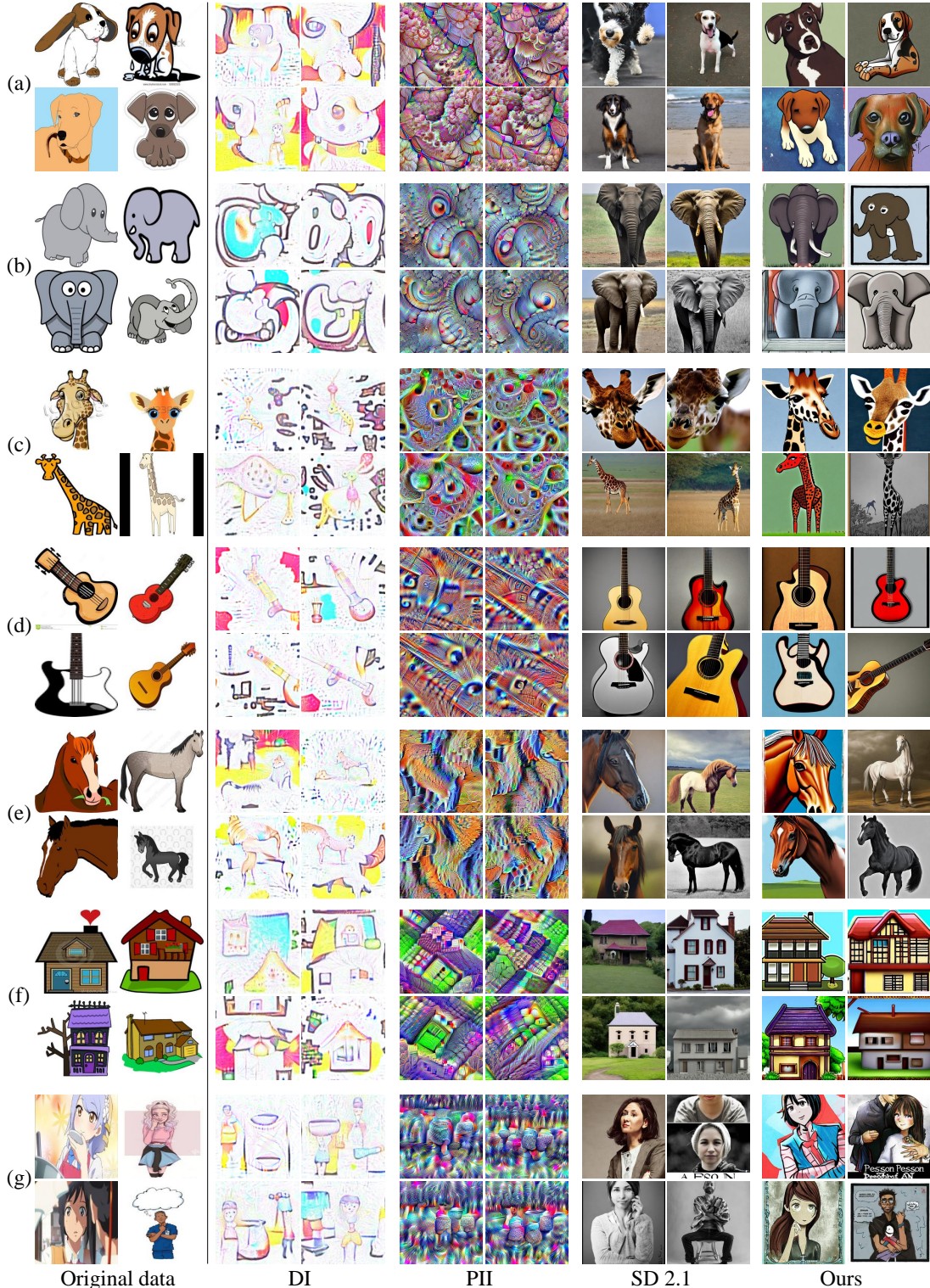

Original data      DI      PII      SD 2.1      Ours

*Figure 14.* Visualization of all class samples in the PACS (*Cartoon*) dataset. (a) through (g) sequentially represent the classes: dog, elephant, giraffe, guitar, horse, house, and person. Our method demonstrates superior capture of target domain and class information compared to baselines (DI, PII) and highlights the domain discrepancy issues that arise when directly applying Stable Diffusion to DFIS.

## Manga

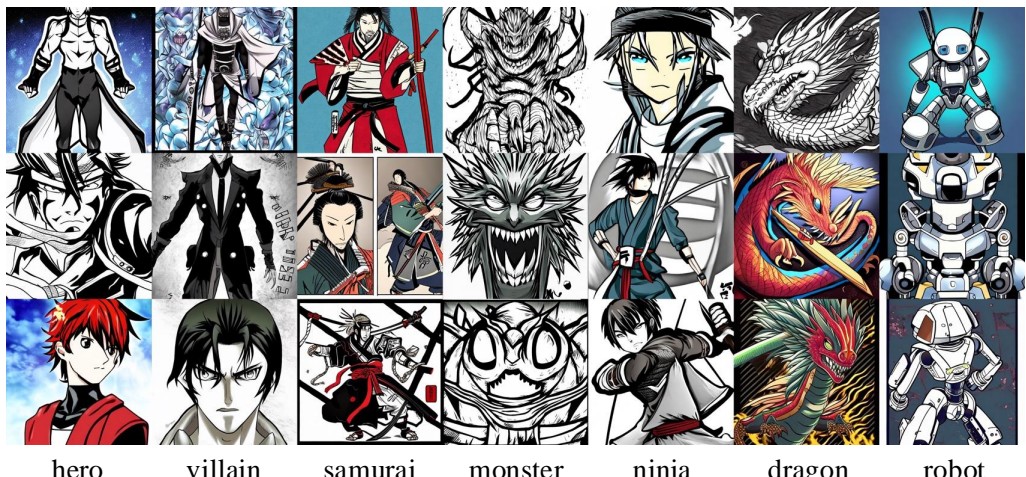

hero    villain    samurai    monster    ninja    dragon    robot

## Caricature

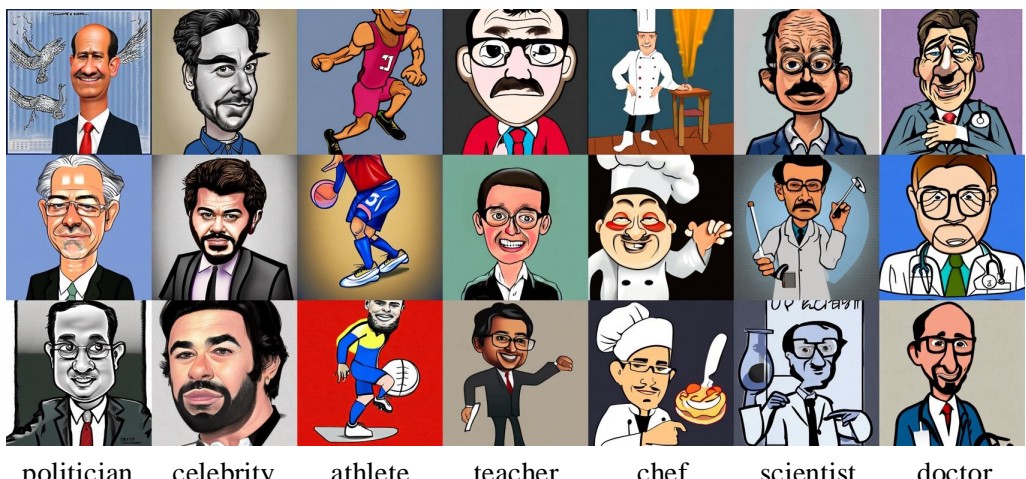

politician    celebrity    athlete    teacher    chef    scientist    doctor

## Sketch

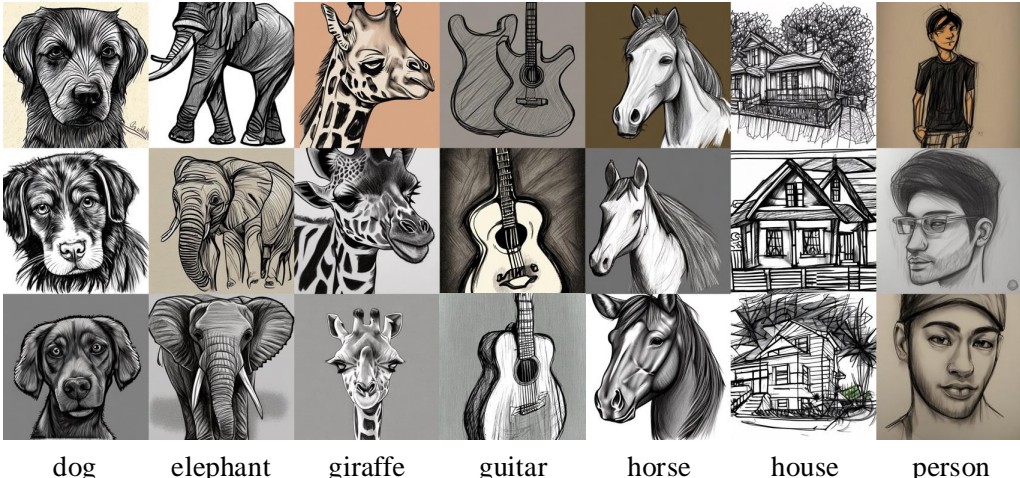

dog    elephant    giraffe    guitar    horse    house    person

*Figure 15.* Synthetic datasets are generated using Style-Aligned *(Original data)*. We treat these synthetic data as the training set and then train a ResNet-34 model. Finally, we invert ResNet-34 pre-trained on a Style-Aligned dataset to generate the images.

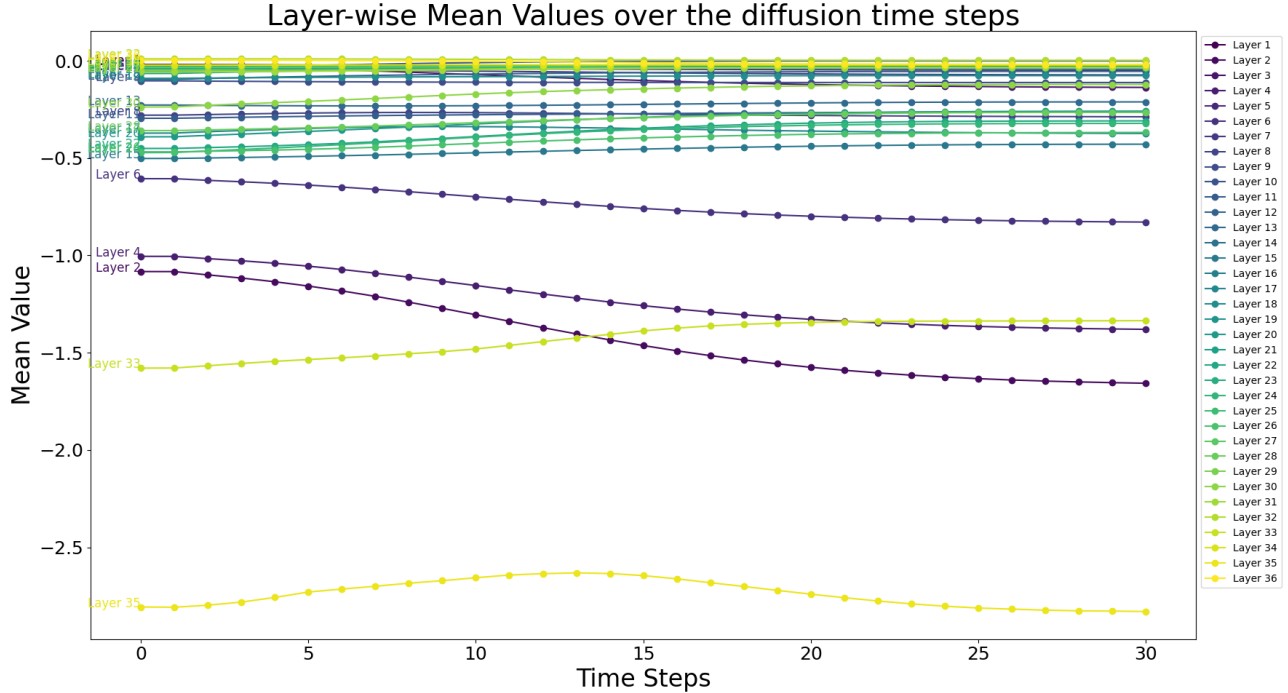

*Figure 16.* Layer-wise mean values

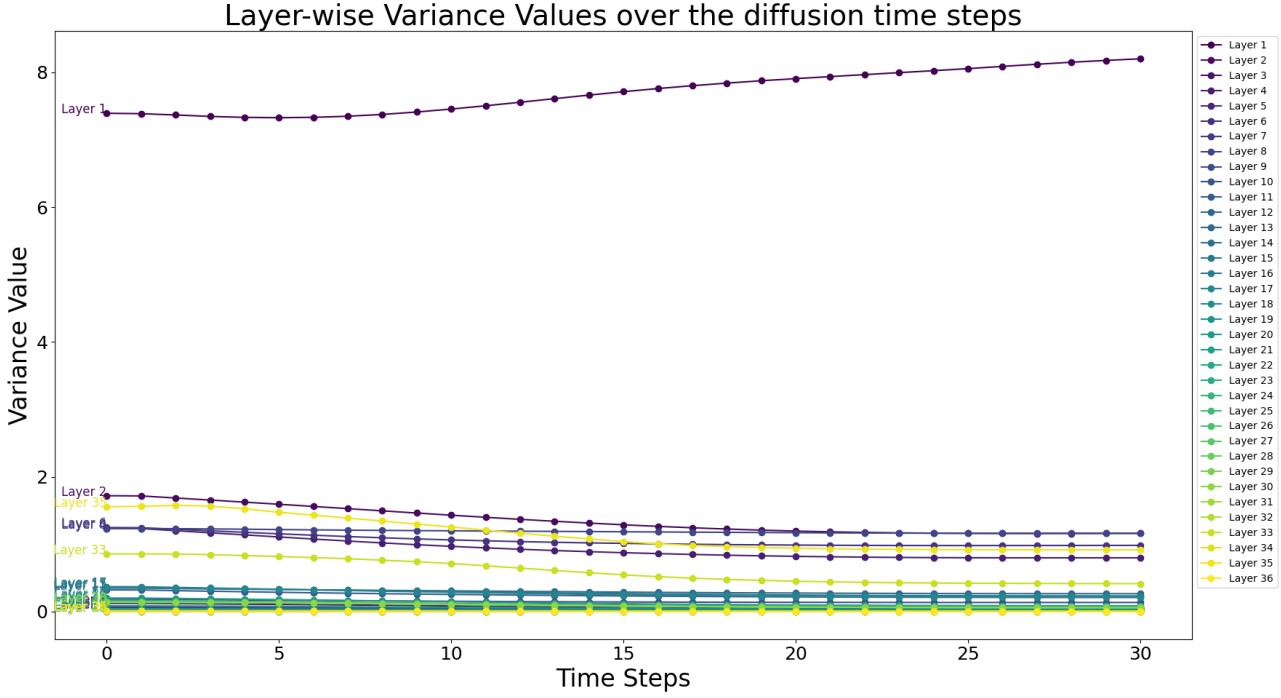

*Figure 17.* Layer-wise variance values

*Figure 18.* Visualization of layer-wise mean and variance over diffusion time steps (DDIM sampler, 30 steps).

1. "a photo of a **degen octopus juggling crypto tokens**"
2. "a photo of a **flex model with VR drip on a digital runway**"
3. "a photo of a **flying submarine gliding over a futuristic harbor**"
4. "a photo of a **ghost kitchen drone delivering byte-snacks**"
5. "a photo of a **posthuman cyborg sipping matcha in a digital cafe**"
6. "a photo of a **robot hummingbird hovering over neon flowers**"
7. "a photo of a **robotic tiger prowling through a cyberpunk alley**"
8. "a photo of a **steampunk robot playing a grand piano in a dimly lit room**"
9. "a photo of an **astronaut riding a giant manta ray in space**"
10. "a photo of a **frosty pixel lion**"

*Figure 19.* Prompt with an unseen class for zero-shot image synthesis. The classes above are unseen by Stable Diffusion 2 (SD2) and were used as prompts to generate images with Stable Diffusion 3 (SD3). For each prompt, we generate 400 images per class, and the resulting images are used as the training dataset for the classifier.

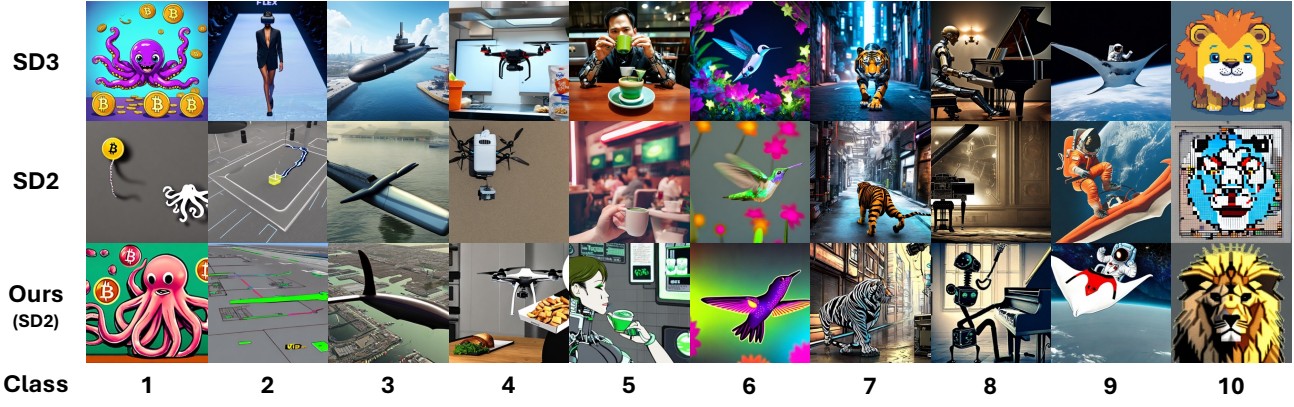

*Figure 20.* Zero-shot image synthesis results for unseen classes. We first find 10 classes described in 19 that Stable Diffusion 2 (SD2) struggles to generate, but Stable Diffusion 3 (SD3) handles well. Using SD3, we generate 400 images per class and then train a ResNet-50 classifier with these images. Using this trained classifier, we apply the DDIS method to synthesize samples that approximate the training set distribution. While the vanilla SD2 (without DDIS) produces samples that deviate significantly from the training set distribution, our proposed method can generate images that closely resemble the original data distribution, even for classes that SD2 has never seen.

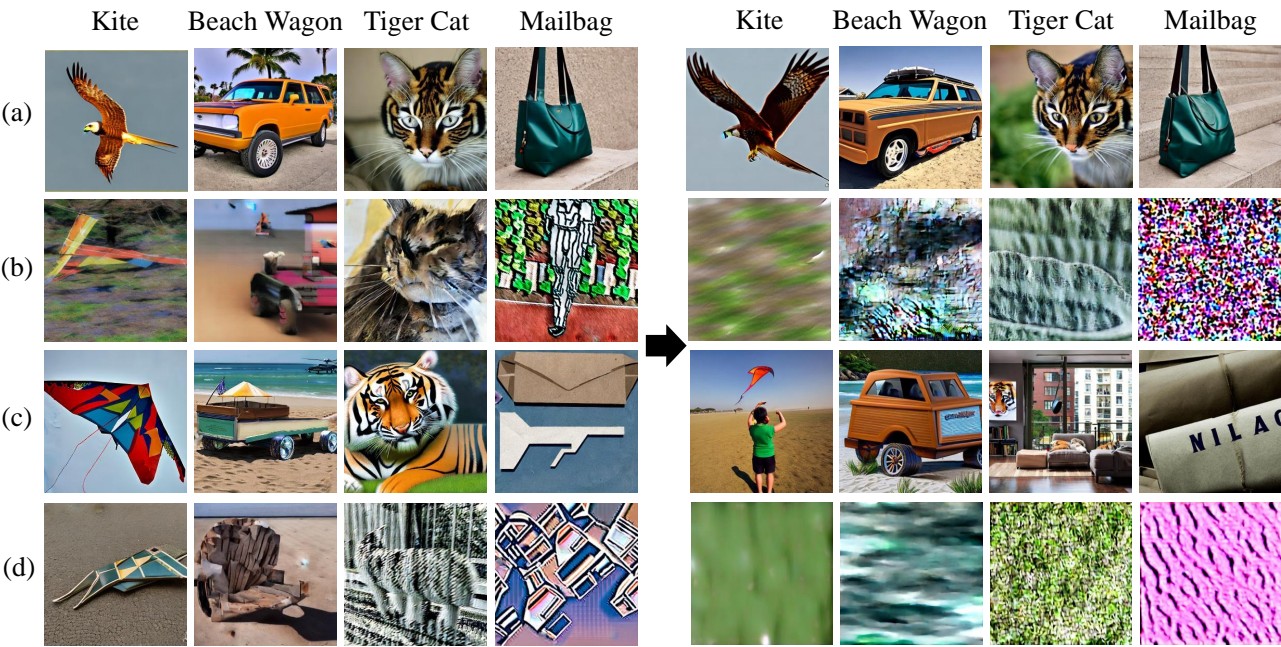

*Figure 21.* Sample visualization for **Design Choice #1** (*Stable Diffusion (SD) Fine-tuning vs. CAT embedding optimization*). Synthetic images for four lexical ambiguous ImageNet-1k classes using (a) our CAT embedding optimization with frozen SD and (b–d) fine-tuning baselines: (b) UNet, (c) text encoder, (d) full SD. For each row, the images on the left of the arrow are generated at the minimum Cross-Entropy loss, and those on the right are after 20 epochs. SD fine-tuning leads to degraded and unstable outputs over time.

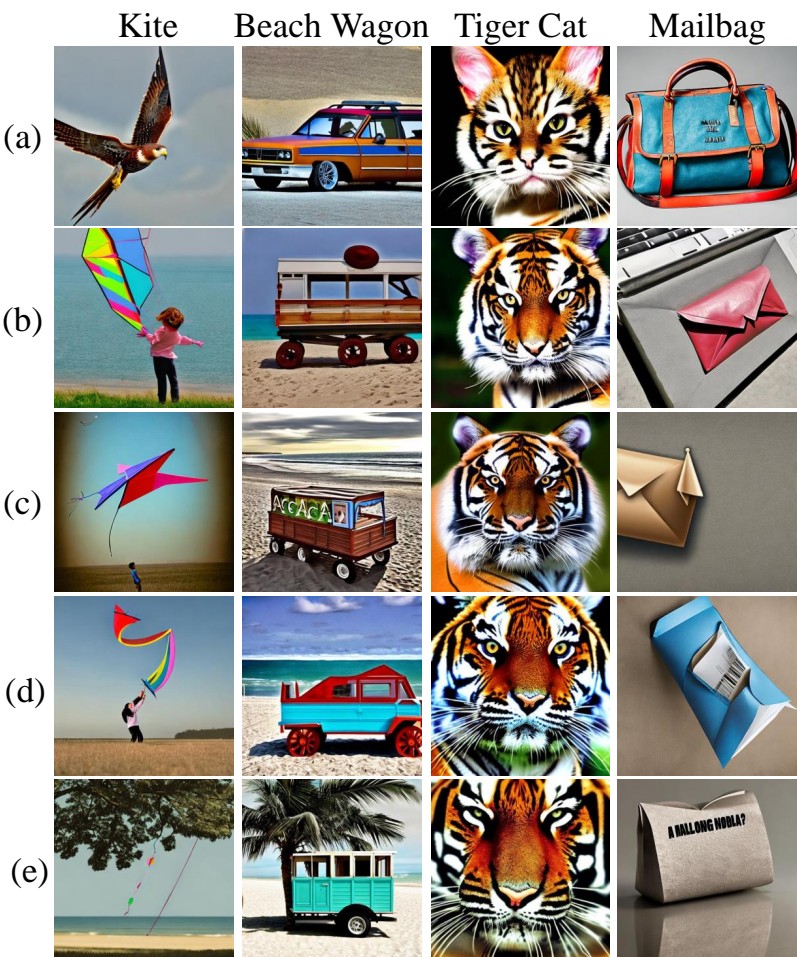

*Figure 22.* Sample visualization for **Design Choice #2** (*Number of CAT tokens*). Synthetic images generated with varying numbers of optimized CAT embeddings on frozen SD: (a) single token (ours), (b-e) from two to five tokens. Image quality degrades as the number of tokens increases and lexical ambiguity persists, indicating that a single token is sufficient for capturing class-specific attributes.

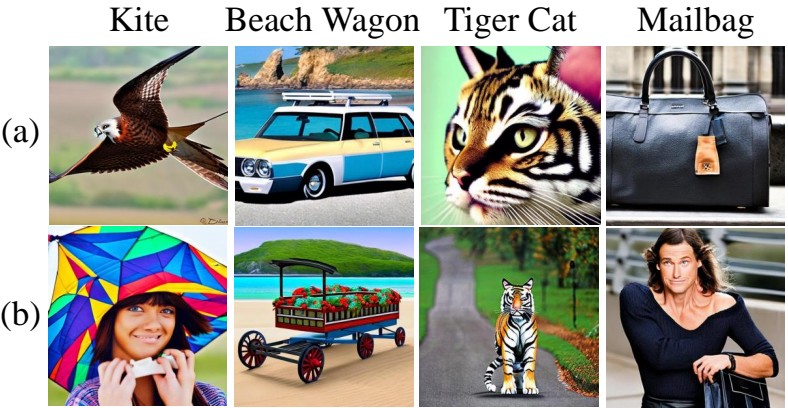

*Figure 23.* Sample visualization for **Design Choice #3** (*Effect of BatchNorm loss on CAT embedding optimization*). (a) CAT embedding optimization with Cross-Entropy loss (ours). (b) CAT optimization with CE and BN loss. BN loss encourages synthetic samples to match overall dataset statistics, often producing mixed-class images and reducing class separability.

