# OpenReview forum: "When Model Knowledge meets Diffusion Model: Diffusion-assisted Data-free Image Synthesis with Alignment of Domain and Class"
_ICML.cc/2025/Conference — ICML 2025 poster_

### Official Review · Reviewer_cfNr · 2025-03-13

**Overall Recommendation:** 2

**Summary:**

Working on the limitations regarding the inefficiency of open-source pre-trained models, which generate samples that significantly deviate from the training data, the authors proposed Diffusion-Assisted Data-Free Image Synthesis (DDIS) in the hope of improving synthetic image quality. The authors first extracted learned distribution knowledge from the given model and used it to guide the diffusion model for image generation. During the sampling process, they introduced domain alignment guidance to align the synthetic data domain with the training data distribution. They experimented with the PACS and ImageNet datasets, validating that the generated samples achieved a better alignment with the training dataset distribution.

**Claims And Evidence:**

Yes, the claims made in the submission are supported by clear and somewhat convincing evidence.

**Essential References Not Discussed:**

The authors covered and discussed SOTA literature related to the domain.

**Experimental Designs Or Analyses:**

I reviewed the complete experimental design and analysis section, including both quantitative and qualitative evaluations, pruning performances, and ablation study.

**Methods And Evaluation Criteria:**

Yes, the proposed methods and evaluation criteria are well-suited to the problem, with relevant benchmark datasets and effective metrics for assessing the network's performance, excluding CLIP-based metrics and classification accuracies.

**Other Comments Or Suggestions:**

The current rating reflects the diffusion-based data-free image synthesis idea, comprehensive experimentations, and ablation studies. I am open to reconsidering the rating if the authors address the raised weaknesses and concerns in the rebuttal.

**Other Strengths And Weaknesses:**

**Strengths**

*S1.* The paper is well-written, with clear pictorial depictions.

*S2.* The idea of data-free image synthesis based on a text-to-image diffusion model is unique and well-motivated.

*S3.* The authors performed comprehensive experiments on image synthesis, knowledge distillation, and pruning.

**Weaknesses**

*W1. Unclear methodological development.* While the idea of integrating a class-token as an embedding in the denoising process is interesting, I found the current methodological development somewhat unclear. How can a learnable class embedding token solve the class alignment problem while keeping all the other network blocks frozen? I believe this might lead to misalignment between the token and the fixed network parameters. While the $v_{Sc}$ ​ might represent an improved class-specific feature, the frozen network may not be able to properly utilize the new token representations.

*W2. Validation of DAG and zero-shot experiments.* While the authors performed ablation studies on the presence of DAG/CAT, they did not thoroughly explore alternative design choices within these components, such as different methods for computing BatchNorm statistics alignment or alternative token optimization strategies. Besides, for a method that modifies tokens to control generation, they should test its generalization to unseen classes or domains, which is completely missing from the experiments.

*W3. Computational complexity.* The authors did not evaluate or compare the computational requirements against baselines. Given the complexity of the dual guidance mechanisms, this is a critical practical consideration that remains unexplored.

*W4. Experimental validation.* While the authors performed DFKD experiments, it lacks direct classification evaluations that would validate its central claims, testing whether standard classifiers correctly identify their generated images, comparing classification accuracy against competing methods, and demonstrating that their synthetic images can effectively train new classifiers from scratch (if possible). I believe these fundamental omissions weaken the claimed efficacy of their CAT contribution despite the model's purported ability to approximate training set distributions.


*Minor point.* In Table 3, the symbols (in the DAG, CAT column) need to be consistent with those in the Method column. For instance, 'SD w/o DAG' should indicate the absence of DAG, so the symbol should appear in the CAT column, not in the DAG column. I might be wrong, please clarify.

**Questions For Authors:**

Please see the weaknesses.

**Relation To Broader Scientific Literature:**

The contributions are related to the broader scientific literature. Additional experiments on downstream tasks using the generated images would further demonstrate the generalizability and robustness of the proposed model.

**Theoretical Claims:**

I believe the paper is more focused on practical applications.

---

> ### Author Rebuttal · Authors · 2025-04-01
>
> Thank you for your detailed comments. We sincerely appreciate your insights and hope to address your concerns.
>
> **W1)** A well-trained generator captures rich knowledge about the training data, enabling it to produce desired images through appropriate combinations of learned features. As a result, many recent studies have explored aligning models to specific tasks using only prompt tuning. The better the pre-trained model, the more effectively it can generalize to unseen combinations within the embedding space. Therefore, optimization within the embedding space alone is often sufficient.
> In our setting, we leverage the strong image prior provided by Stable Diffusion. This allows even a simple prompt such as “a photo of [class]” to effectively generate class-representative images (see “SD 2.1” in Fig. 2–5).  However, the reason we utilize the CAT token is that the generated class is not perfectly aligned with the classifier’s learned class distribution (see Fig. 5, third column). For example, in ambiguous cases such as “kite,” which can refer to both a raptor and a toy, or for classes that require fine-grained details like “tiger cat,” the use of a Class token alone is sufficient to achieve closer alignment with the original dataset’s learned class distribution (see Fig. 5, fourth column).
>
> **W2-1) Validation of DAG**: To align BN statistics, we design a guidance mechanism within the diffusion sampling process. To address this concern of our DAG design, we conduct experiments using alternative design choices for BN statistics alignment. Specifically, instead of applying guidance during the sampling process, we optimize the CAT embedding directly using Equation (9) with the Cross-Entropy loss as the objective function. The results shown in Figure 2 of the attached link demonstrate the outcome when CAT token optimization is performed without DAG, where both cross-entropy loss and BN loss are optimized during training. In this case, although each PACS-art sample captures certain domain characteristics, the influence toward the target class becomes weaker. As a result, the generated images tend to lose critical target class attributes, leading to degraded class-specific representations. Furthermore, BN loss encourages the synthetic images to follow the averaged running statistics over the entire training set, which can lead to the generation of images that contain mixed information from multiple classes.
>   In contrast, our proposed DAG performs BN statistic alignment only during the diffusion sampling process, while the CAT embedding is optimized solely using cross-entropy loss. This separation allows the CAT embedding to focus on capturing class semantics while the sampling process handles domain characteristics. As a result, our method effectively approximates both the class and domain properties of the original training data.
>
> **W2-2) Zero-shot image synthesis** We conduct a zero-shot image synthesis experiment to evaluate whether Stable Diffusion can effectively generate images for unseen classes. First, using Stable Diffusion V3, we select 10 classes that the Stable Diffusion 2.1 model used in our study has never encountered and generate images for those classes with SD V3. Then, we train a ResNet-50 model using the dataset of these 10 classes. Finally, we utilize Stable Diffusion 2.1 to synthesize the images used for training ResNet-50 with the our proposed DDIS. Surprisingly, as shown in Figure 4 in the link, although Stable Diffusion 2.1 struggles with these classes, the CAT embedding can capture the class attributes from the training set and generate images with a distribution similar to that of the original dataset. (Please see this link: https://drive.google.com/file/d/1S2Qt0XSgMH2pOzdzsNekhlyP-bLBY5kt/view?usp=sharing)
>
> **W3)** Please refer to our response to Reviewer #e4Ls’s **W1)** for a detailed explanation.
>
> | Method | DI | PlI | SD|Ours  |
> |:------------:|:-------:|:-----------:|:-----:|:-----:|
> | ImageNet-1k | 32.11\%       | 27.37\%        | 64.31\%  |76.61\%|
> | PACS-art | 30.24\%       | 22.85\%        | 47.51\%  |54.29\%|
> | PACS-cartoon | 34.56\%       | 22.15\%        | 49.79\%  |67.37\%|
>
> **W4)** To assess whether our generated images are distinguishable by standard classifiers, we conduct an additional experiment where we pass 100,000 synthetic images through a ResNet-34 pretrained on ImageNet-1k and report the average top-1 classification confidence for each method. As shown in the above Table, our method achieves the highest average confidence across 1000 classes, indicating that our synthetic samples are more consistent with the model's learned distribution. These results further support the effectiveness of CAT embedding in generating class-consistent and classifier-recognizable images. We will include this analysis in the future to reinforce the practical impact of our method.

---

> > ### Comment · Reviewer_cfNr · 2025-04-06
> >
> > Thank you for the detailed rebuttal. While the authors provide clarifications and additional experiments, I find that the methodological development around CAT embedding with frozen networks remains unclear, broader ablations on design alternatives are still lacking, and validation through classification-based evaluations is limited. That being said, I will maintain my original score.

---

> > > ### Author Response · Authors · 2025-04-09
> > >
> > > We would like to express our sincere gratitude for your insightful comments. We genuinely hope that our responses have sufficiently addressed the concerns raised.
> > >
> > > **1) Further Analysis of CAT Embedding Optimization with Frozen SD Networks**
> > >
> > > To evaluate the effectiveness of our CAT embedding optimization, we conducted ablation studies on three design choices, synthesizing four lexically ambiguous ImageNet-1K classes: Kite (21), Tiger Cat (282), Beach Wagon (436), and Mail Bag (636) like Fig.5 in the main paper. We compared confidence scores and visual quality under these settings (see the link: https://drive.google.com/file/d/1WkehcgRb--H8gGkI_o6LqfrnKAs89vMl/view?usp=drive_link and the tables below).
> > >
> > > * ***Design Choice 1: Partial or Full Fine-tuning of SD***
> > >
> > > | Method | C-21 | C-282 | C-436 | C-636 | Avg.Confidence  |
> > > |--------|------|-------|-------|-------|------|
> > > | (a) Ours | 94.12 | 65.94 | 85.73 | 67.66 | 78.36 |
> > > | (b) SD UNet | 0.77 | 0.67 | 2.47 | 1.53 | 1.36 |
> > > | (c) SD Text-Encoder | 0.03 | 21.25 | 17.35 | 2.57 | 10.30 |
> > > | (d) SD Full | 0.26 | 0.40 | 0.01 | 0.01 | 0.17 |
> > >
> > > We tested three fine-tuning configurations for Stable Diffusion (SD): (b) UNet only, (c) text encoder only, and (d) Full fine-tuning, using the Cross-Entropy (CE) loss used for CAT optimization. As shown in Fig. 1 (attached link) and the table above, fine-tuned SD produces distorted, low-confidence images, failing to generate class-aligned images. It suggests that the SD fine-tuning in Data-Free Image Synthesis (DFIS) disrupts the prior knowledge within the SD, degrading image quality.
> > >
> > > Generally, SD is fine-tuned using real data to adapt to specific domains or styles, but in DFIS, the lack of real images leads to unstable training (right side of Fig. 1 (attached link)). Moreover, per-class SD fine-tuning leads to individual SD networks, increasing computational cost. In contrast, our method freezes SD and optimizes a single token, preserving SD's image prior while efficiently generating high-quality, class-aligned images.
> > >
> > > * ***Design Choice 2: Optimizing Multiple Tokens Embeddings***
> > >
> > > | Method | C-21 | C-282 | C-436 | C-636 | Avg. Confidence |
> > > |--------|------|-------|-------|-------|------|
> > > | 1 Token (Ours) | 94.12 | 65.94 | 85.73 | 67.66 | 78.36 |
> > > | 2 Tokens | 0.01 | 30.19 | 5.04 | 4.10 | 9.83 |
> > > | 3 Tokens | 0.01 | 30.15 | 5.08 | 3.19 | 9.60 |
> > > | 4 Tokens | 0.01 | 30.17 | 5.07 | 3.59 | 9.71 |
> > > | 5 Tokens | 0.01 | 29.14 | 1.32 | 3.54 | 8.50 |
> > >
> > > We explore whether multiple CAT embeddings improve class expressivity by optimizing embeddings for one to five tokens. As shown in Figure 2 (attached link) and the table above, performance drops with more tokens. Lexical ambiguity persists, and confidence scores drop. In data-free settings with simple prompts (e.g., "A $S_c$ {class}"), more tokens amplify the effect of randomly initialized embeddings, hindering desired class-aligned image generation. Therefore, a single token is sufficient and more effective for encoding class information in DFIS.
> > >
> > > * ***Design Choice 3: Adding BatchNorm loss on CAT embedding optimization***
> > >
> > > | Method        | C-21 | C-282 | C-436 | C-636 | Avg. Confidence |
> > > |---------------|------|-------|-------|-------|------|
> > > | Ours| 94.12 | 65.94 | 85.73 | 67.66 | 78.36 |
> > > | CAT optim. w/BN Loss   | 0.01  | 26.15 | 8.26  | 0.03  | 8.61  |
> > >
> > > We tested optimizing CAT embedding with BatchNorm (BN) loss alongside vanilla CE loss. As shown in Fig. 3(attached link) and the above table, BN loss negatively affects capturing class semantics by inducing synthetic images toward the averaged statistics of the entire dataset, which causes class mixing and reduces separability. Therefore, optimizing CAT embedding with CE loss alone effectively captures class-specific attributes.
> > >
> > > * ***Conclusion***
> > >
> > > Existing studies [1–3] validate that desired concepts or complex information can be encoded by optimizing only token embeddings on frozen SD. Building on this, we adapt the idea to DFIS and demonstrate that a single CAT token effectively captures class semantics while preserving SD’s priors. Across all design choices, our approach consistently outperforms alternatives, highlighting its effectiveness for DFIS.
> > >
> > > [1] Visual Instruction Inversion, NeurIPS 2023
> > >
> > > [2] ReVersion, SIGGRAPH 2024
> > >
> > > [3] Open-Vocabulary Attention Maps, CVPR 2024
> > >
> > > **2) Classification-Based Evaluation**
> > >
> > > We evaluate the top-1 accuracy of 100K synthetic samples using DI (RN-34), PII (RN-34), SD, and our method (RN-34) on *53 classifiers* pre-trained on ImageNet-1k. As shown in Table 1 (attached link), DDIS outperforms PII, DI, and SD by  49.27%, 44.53%, and 6.37% on average. Despite using ResNet-34 for generation, DDIS generalizes well, aligning closely with the training distribution.
> > >
> > > We plan to include the above experimental results in the future. If there are any remaining concerns, we would greatly appreciate your further comments. It has been an honor to discuss our work with you in this rebuttal process.

---

### Official Review · Reviewer_UzMa · 2025-03-13

**Overall Recommendation:** 3

**Summary:**

- This paper proposes DDIS, a data-free image synthesis method that leverages a pre‐trained text-to‐image diffusion model as an image prior.
- It introduces two key components: Domain Alignment Guidance (DAG), which uses the classifier’s batch normalization statistics to steer the diffusion sampling toward the original data domain, and a Class Alignment Token (CAT) that learns to capture class‐specific features, thereby resolving lexical ambiguities.
- The method is shown to generate images that more closely reflect the unknown training data distribution and to improve downstream tasks such as data‐free knowledge distillation.

**Claims And Evidence:**

- The authors claim that integrating diffusion guidance with DAG and CAT significantly improves the approximation of training set distribution, as evidenced by improved FID, precision/recall, and better performance in data‐free distillation.
- Quantitative results on datasets like PACS and ImageNet support these claims, although the discussion on the limitations of prior DFIS methods could be more detailed.
- Overall, the evidence supports the claims.

**Essential References Not Discussed:**

[1] Mokady, R., Hertz, A., Aberman, K., Pritch, Y. and Cohen-Or, D., 2023. Null-text inversion for editing real images using guided diffusion models. In Proceedings of the IEEE/CVF conference on computer vision and pattern recognition (pp. 6038-6047).

[2] Gal, R., Alaluf, Y., Atzmon, Y., Patashnik, O., Bermano, A.H., Chechik, G. and Cohen-Or, D., 2022. An image is worth one word: Personalizing text-to-image generation using textual inversion. arXiv preprint arXiv:2208.01618.

**Ethical Review Concerns:**

The primary goal of the proposed method is to generate images that approximate the inaccessable training data distribution. The authors have mentioned that ‘training datasets are often inaccessible due to various constraints such as data privacy concerns and copyright issues.’ (l.12-13), indicating that recovering training image in a close enough manner may be broken the privacy constrain. However, I didn’t see a dedicated discussion regarding the Ethics Impact; in fact, the whole Broader Impact section required was missing. Therefore, I raise an ethics review for this submission.

**Ethical Review Flag:**

Flag this paper for an ethics review.

**Ethics Expertise Needed:**

["Privacy and Security"]

**Experimental Designs Or Analyses:**

- The experiments are comprehensive, evaluating performance on multiple datasets and downstream tasks.
- l.365-367 ‘The experimental setup for knowledge distillation is based on the protocol outlined in (Li et al., 2023).’, please brifely explain the experiment setup rather than redirecting the reader to the reference work.

**Methods And Evaluation Criteria:**

- Conceptually, the Domain Alignment Guidance (DAG) proposed in this work is extending the famous Null-text Inversion [1] into a full distribution setting, through batchify optimisation equipted with batch normalization and runnign statistics. It was ok to convert the existing method to a different problem setting; however, I was surprised to see that such a closely related work was completely missing through out the paper.
- The relationship of the 𝐂𝐥𝐚𝐬𝐬 𝐀𝐥𝐢𝐠𝐧𝐦𝐞𝐧𝐭 𝐓𝐨𝐤𝐞𝐧 (𝐂𝐀𝐓) to Textual Inversion [2] should also be discussed.

**Other Comments Or Suggestions:**

n/a

**Other Strengths And Weaknesses:**

- Strengths: The integration of a diffusion model with BN-based domain guidance and class-specific tokens is innovative and produces strong empirical improvements in synthetic images that approximate unseen training data distribution.
- Weaknesses: There is no major technique issue with the submission, but I hold my recommendation until an Ethical Review suggested below.

**Questions For Authors:**

See above.

**Relation To Broader Scientific Literature:**

See below the ‘Essential References Not Discussed’ section

**Theoretical Claims:**

There is no major issue with the theoretical claims, the finding that hacking BN statistics of pre-trained models and combined with image generation can reproduce training distribution is interesting.

---

> ### Author Rebuttal · Authors · 2025-04-01
>
> Thank you for providing meaningful feedback. We appreciate your thoughtful consideration. We have
>  tried our best to provide responses to address your comments.
>
> **Claims and Evidence**
>
> Please refer to our response **W1)** to Reviewer #83BW for a detailed explanation. As a result of supporting the above limitation of previous works, prior methods fail to approximate the original data distribution, resulting in low-quality synthesized images shown in Figure 2-5 in the main paper. This mismatch also negatively impacts performance in downstream applications, shown in Table 2 and Figure 6 in the main paper, further highlighting the practical limitations of existing approaches.
>
> **Methods and Evaluation Criteria**
>
> Thank you for providing valuable discussion points regarding related works. We have carefully reviewed all the papers you mentioned and have identified the following key differences between our work and theirs.
>
> |                | Textual Inversion                                                | Null Text Inversion                                                              | Ours                                                    |
> |----------------|------------------------------------------------------------------|----------------------------------------------------------------------------------|---------------------------------------------------------|
> | **Task**           | personalized image generation                                    | image editing                                                                    | data-free image synthesis                               |
> | **Input**          | few (3-5) source images                                            | single source image                                                              | pre-trained classifier                                  |
> | **Output**         | personalized image                                               | edited image                                                                     | image that follows classifier's training data distribution |
>
> *1)Null-Text Inversion*: We would like to first clarify the differences between Null-Text-Inversion(NTI) and the problem setting of our research(DAG). NTI is primarily focused on effective image editing by performing DDIM inversion to obtain z_T​ from a given input image. In contrast, our goal is to synthesize data that approximates the distribution of the training dataset learned by a classifier—without requiring any input image. We start from random noise and leverage the classifier’s knowledge(BatchNorm statistics) as guidance. In other words, while NTI requires a source image as input and produces an edited version of that image as output, our method only requires an arbitrary classifier. To ensure clarity, we will explicitly detail these differences in the revised version of the paper.
>
> *2)Textual Inversion (TI)*: TI and DDIS both optimize a pseudo-word token embedding, but with different goals. TI personalizes text-to-image models by learning an embedding from a few reference images to capture a visual concept, guiding image generation accordingly. In contrast, DDIS generates data approximating a classifier’s training distribution without reference images, optimizing the embedding via cross-entropy loss between classifier predictions and target labels to steer generation toward the target class. While TI enables controlled generation, our method leverages a fixed diffusion model as a strong prior to constrain the image search space in data-free synthesis, yielding realistic, class-aligned samples.
>
>
> We also appreciate your feedback regarding the ethical review. We will provide the following discussion and incorporate it into the revised version of the paper.
>
> **Ethical Review**
>
> This work aims to enhance the utility of pre-trained models by generating synthetic data in scenarios where access to the original data is unavailable. We acknowledge that data-free image synthesis approaches, designed to recover data following the training dataset’s distribution, can raise concerns regarding privacy leakage and other ethical issues. However, our objective is not to reconstruct individual instances or facilitate unauthorized data access but rather to synthesize a surrogate dataset that can be effectively utilized for data-free applications such as data-free knowledge distillation or data-free pruning. Specifically, our method guides the data generation process by leveraging the running statistics from classifiers, which represent the averaged information of the entire dataset. This design inherently prevents the inclusion of individual instance details in the generated images, making it extremely difficult to recover any specific sample (see Figs. 2–5 in the main paper). We hope that this work contributes to the responsible development of data-free techniques for scenarios in which data sharing is limited or infeasible.

---

### Official Review · Reviewer_e4Ls · 2025-03-17

**Overall Recommendation:** 3

**Summary:**

This paper focuses on DataFree Image Synthesis and claims that the existing methods usually produce samples that deviate significantly from the training data distribution. To address this problem, the authors aim to leverage a text-to-image diffusion model to extract knowledge about the learned distribution. To achieve this, the authors propose Domain Alignment Guidance and Class Alignment Token module. The experiments show that the proposed method can achieve state-of-the-art performance in data-free applications.

**Claims And Evidence:**

The claim, i.e., all the existing methods usually produce samples that deviate significantly from the training data distribution, needs evidences.

**Essential References Not Discussed:**

N/A

**Experimental Designs Or Analyses:**

I check the validity of any experimental designs.

**Methods And Evaluation Criteria:**

Yes

**Other Comments Or Suggestions:**

N/A

**Other Strengths And Weaknesses:**

Strengths
1. The idea of leveraging a text-to-image diffusion model as a powerful image prior is reasonable.
2. State-of-the-art performance in data-free applications is achieved.

Weaknesses
1. The Computation Overhead is still heavy.
2. The claim, i.e., all the existing methods usually produce samples that deviate significantly from the training data distribution, needs evidence.
3. The batch normalization (BN) has been proved that it can reflect domain-related knowledge. The DAG starts from this and leverages CFG  to achieve Domain Alignment Guidance. The key differences over the existing methods are still not clear.

**Questions For Authors:**

Please see the weakness.

**Relation To Broader Scientific Literature:**

The key contribution of leveraging a text-to-image diffusion model as a powerful image prior can related to the broader scientific literature that diffusion models can serve as plug-and-play priors.

**Theoretical Claims:**

No Theoretical Claims

---

> ### Author Rebuttal · Authors · 2025-04-01
>
> We would like to express our gratitude for your thoughtful comments and valuable feedback. We understand your concerns, and hope the following response addresses them.
>
> | Method                                     | DeepInversion | PlugInInversion | Ours  |
> |:------------------------------------------:|:-------------:|:---------------:|:-----:|
> | Total Iteration for 100k samples synthesis (ImageNet-1k) | 80,000K       | 1,120K          | 30K   |
> | Times per 1 iteration (sec)                | 0.83          | 0.79            | 15.17 |
> | Total training cost (hours)                | 18444         | 245             | 126   |
>
> **W1)** Compared to existing DFIS methods, our DDIS offers significant improvements in efficiency and cost-effectiveness for large-scale dataset generation on ImageNet-1k while maintaining high image quality. To illustrate, we compare the number of optimization iterations required to synthesize 100,000 ImageNet-1k images. All experiments were conducted using a single RTX 4090 GPU:
>
> * **DeepInversion (DI)**: DI requires iterative optimization for each batch of images. With an optimal batch size of 250 on a single RTX 3090, approximately 20,000 iterations are needed per batch. To generate 100,000 images, DI must repeat this process 400 times, resulting in a total of 800,000K iterations.
>
> * **PlugInInversion (PII)**: PII similarly optimizes a randomly initialized input but across 7 progressive upsampling stages (from 7×7 to 224×224), with 400 iterations per stage. Thus, generating 100,000 images with a batch size of 250 results in around 1,120K total iterations.
>
> * **DDIS (Ours)**: DDIS optimizes only class-wise CAT embedding vectors in a relatively low-dimensional space (1×784). For ImageNet-1k with 1000 classes, we perform just 30 iterations per class, totaling only 3,000 iterations. Once a CAT embedding is found, we can synthesize 100,000 images simply by sampling latent vectors—without additional training or optimization. CAT optimization takes slightly longer per iteration than prior works, but the total required iterations are significantly lower, making the overall process more efficient.
>
> **W2)** As you said, comparing data distributions is inherently challenging. Generally, in the image synthesis tasks, Is, FID, Precision, and Recall are widely used to quantify fidelity and diversity, respectively, providing a comprehensive view of how closely the synthetic data approximates the real training distribution. The metric results presented in Table 1 reflect this distributional gap between the real and synthetic data. Our proposed method, DDIS, consistently outperforms prior approaches across all metrics (IS, FID, precision, and recall), indicating strong alignment with the true data distribution. This is further supported by the results in Table 2 and Figure 6, where using our synthetic data in place of the original training set yields the best performance in applications such as DFKD and pruning. We would also like to say that Reviewer #UzMa, in Claims and Evidence #1, acknowledged these results as strong evidence supporting the effectiveness of our approach.
>
> **W3)** In our view, the proposed method, DAG, is conceptually more aligned with Classifier Guidance than with Classifier-Free Guidance. DAG guides the diffusion sampling process by leveraging the running statistics stored in a classifier, which closely resembles the mechanism of Classifier Guidance. In traditional Classifier Guidance, It requires $t$ additional classifiers trained on noisy latents $z_t$ at different time step t, making it incompatible with a standard pretrained classifier. In contrast, DAG makes use of batch normalization statistics instead of classifier outputs, allowing it to work with general classifiers that are trained on $z_0​. Furthermore, A key distinction lies in the role each method plays: DAG provides guidance based on the training set domain, encouraging the generated samples to follow the underlying distribution of the original data. On the other hand, CG is typically used to guide an unconditioned diffusion model to generate images belonging to a specific target class.
>
>  Classifier-Free Guidance (CFG), by contrast, does not rely on a separate classifier and instead offers text-conditioned guidance to ensure prompt fidelity. Importantly, DAG and CFG can be used together, as their mechanisms are orthogonal and complementary.
>
> In summary, DAG shares the classifier-based foundation of CG but distinguishes itself by using the gradient of the BN loss and focusing on aligning samples with the overall training distribution, rather than generating class-specific outputs.

---

### Official Review · Reviewer_83BW · 2025-03-18

**Overall Recommendation:** 3

**Summary:**

This paper proposed a novel Diffusion-assisted Data-free Image Synthesis method designed to improve the quality of images generated without access to training data. Traditional methods struggle to approximate the original data distribution due to the absence of natural image priors. DDIS overcomes this by leveraging a text-to-image diffusion model as a strong prior, extracting knowledge from a pre-trained model to guide synthesis. The approach introduces Domain Alignment Guidance (DAG) to align synthetic images with the training data and a Class Alignment Token (CAT) for capturing class-specific attributes. Experiments on PACS and ImageNet confirm DDIS’s superiority over existing methods.

##update after rebuttal
The authors have addressed most of my concerns satisfactorily, I am therefore updating my recommendation to a weak accept.

**Claims And Evidence:**

1. The related work section states that finding an optimal sample that closely approximates the training distribution within a vast search space remains a significant challenge. However, it should explicitly clarify the specific limitations of prior work in addressing this issue and how this paper overcomes them. Identifying gaps in existing methods and demonstrating the proposed approach’s advantages would strengthen the argument.

2. This paper claims to be the first to successfully generate samples across various domains beyond photo datasets. However, this claim lacks a direct comparison with data-free knowledge distillation methods, some of which also focus on generating training samples. Without such a comparison, the claim remains unsubstantiated.

**Essential References Not Discussed:**

This paper lacks discussion on existing data-free knowledge distillation methods in the related work section. Given that the primary objective of data-free image synthesis in this study is to facilitate downstream tasks such as knowledge distillation, it is crucial to position the proposed approach within the broader context of prior research.

[1] Gongfan Fang, Jie Song, Xinchao Wang, Chengchao Shen, Xingen Wang,
and Mingli Song. Contrastive model inversion for data-free knowledge dis-
tillation. arXiv preprint arXiv:2105.08584, 2021.

[2] Shikang Yu, Jiachen Chen, Hu Han, and Shuqiang Jiang. Data-free knowl-
edge distillation via feature exchange and activation region constraint. In
Proceedings of the IEEE/CVF Conference on Computer Vision and Pattern
Recognition, pages 24266–24275, 2023.

[3] Minh-Tuan Tran, Trung Le, Xuan-May Le, Mehrtash Harandi, Quan Hung
Tran, and Dinh Phung. Nayer: Noisy layer data generation for efficient and
effective data-free knowledge distillation. In Proceedings of the IEEE/CVF
Conference on Computer Vision and Pattern Recognition, pages 23860–
23869, 2024.

**Experimental Designs Or Analyses:**

1. The logical correctness of Algorithm 1, particularly within the for loop, appears questionable. The usage of the final guided latent after the DAG process in the subsequent iteration is unclear.

2. Considering the high computational cost of diffusion models, how does DDIS compare in efficiency with other DFIS methods? Additionally, the optimization of the class alignment token seems time-consuming, yet its implementation details are not addressed.

**Methods And Evaluation Criteria:**

1. The methods should be compared with recent data-free knowledge distillation approaches, as listed in the "Essential References Not Discussed" section, to ensure a comprehensive evaluation.

2. The paper states that it does not compare performance with NaturalInversion because NI focuses on small-scale datasets. However, the rationale for this exclusion is unclear.

3. The proposed method leverages batch normalization (BN) statistics, which limits the proposed method's generalization to other model architectures.

**Other Comments Or Suggestions:**

1. In the phrase "Expressively, since the Text conditioned", the word "Text" should be in lowercase ("text") to maintain consistency in terminology unless it specifically refers to a defined term or concept within the paper.

2. Comma Placement within Formula (Line 247, Left Column, Page 5): It would be more appropriately included within the notation of formula (10).


3. The phrase "=0" appearing at the last line of both algorithms seems redundant.

4. The sentence "We evaluate image quality of synthetic 10k ImageNet-1k, 2.8k PACS, and 1.4k Style-Aligned samples." (Line 345, Right Column, Page 7) requires verification. Specifically, the phrasing "synthetic 10k ImageNet-1k" may lack clarity.

5. The abbreviation "DE methods" in the caption of Figure 6 and Table 2 should be explicitly defined to enhance clarity and avoid ambiguity. If it refers to a known methodology, consider providing a brief explanation or reference upon its first occurrence.

6. There appears to be a repetition in these lines (Lines 424-426, Right Column, Page 8). A careful review is needed.

**Other Strengths And Weaknesses:**

The strengths and weaknesses have been presented above.

**Questions For Authors:**

1. The related work section states that finding an optimal sample approximating the training distribution within a vast search space is a significant challenge. However, the paper does not clearly specify the limitations of prior methods in addressing this issue. Could the authors explicitly outline these limitations and how the approach overcomes them? A clear comparison would help clarify the novelty and effectiveness of your method.

2. The paper claims to be the first to generate samples across various domains beyond photo datasets. However, this claim lacks a direct comparison with existing DFKD methods, some of which also generate training samples. How does the proposed method compare to recent DFKD approaches, such as those listed in the “Essential References Not Discussed” section? A comparative analysis would help substantiate the uniqueness of the contributions.

3. The paper states that it does not compare performance with NI because NI focuses on small-scale datasets. Could you clarify why this exclusion is justified? Would NI not provide at least some insights into small-scale performance, or could an adapted comparison be made? Including such a discussion would strengthen the completeness of the evaluation.


4. In Algorithm 1, the logical correctness within the for loop is unclear, particularly regarding the use of the final guided latent after the DAG process in the subsequent iteration.

5. Given the high computational cost of diffusion models, how does DDIS compare in efficiency with other Data-Free Image Synthesis (DFIS) methods? Additionally, the optimization of the class alignment token seems time-consuming, yet implementation details are not provided. It would be better to elaborate on its efficiency and potential optimizations?

6. The related work section lacks discussion about existing DFKD methods.

**Relation To Broader Scientific Literature:**

The design of domain adaptive guidance might be related to the broader scientific literature.

**Theoretical Claims:**

This paper does not present any formal proofs or theoretical claims.

---

> ### Author Rebuttal · Authors · 2025-04-01
>
> We would like to express our gratitude for your thoughtful comments and valuable feedback. Our response to the weakness is as follows.
> |       Method (C10)      |   IS   |   FID   | Precision | Recall  | KD Result (\%) |
> |:------------------:|:------:|:-------:|:---------:|:-------:|:---------:|
> | [1] CMI            |  2.78  | 178.42  |  0.5048   | 0.0745  |   94.84   |
> | [2] SpaceShipNet   |  3.02  | 165.40  |  0.5117   | 0.0146  |   95.39   |
> | [3] NAYER          |  2.53  | 186.51  |  0.4847   | 0.0062  |   95.21   |
> | Ours               |  5.31  |  38.93  |  0.7280   | 0.5192  |   91.44   |
>
> |     Methods (C100)     |  IS   |  FID  |   P    |   R    | KD Result |
> |:---------------:|:-----:|:-----:|:------:|:------:|:---------:|
> | [1] CMI         | 3.94  | 167.29| 0.4686 | 0.0114 |   77.04   |
> | [2] ShipNet     | 4.52  | 158.73| 0.4893 | 0.0132 |   77.41   |
> | [3] NAYER       | 3.87  | 157.28| 0.4984 | 0.0105 |   77.54   |
> | DDIS (Ours)     | 7.43  | 28.66 | 0.7945 | 0.5355 |   72.57   |
>
>
> |      CIFAR-10      |  50%  |  60%  |  70%  |  80%  |  90%  |
> |:-----------------:|:-----:|:-----:|:-----:|:-----:|:-----:|
> | [1] CMI           | 76.15 | 70.41 | 61.73 | 52.56 | 48.94 |
> | [2] SpaceShipNet       | 78.61 | 72.26 | 62.41 | 53.70 | 51.81 |
> | [3] NAYER         | 77.66 | 74.31 | 60.51 | 55.72 | 52.78 |
> | DDIS (Ours)       | 78.82 | 74.51 | 64.72 | 59.69 | 57.81 |
>
> |     CIFAR-100    |  50%  |  60%  |  70%  |  80%  |  90%  |
> |:---------------:|:-----:|:-----:|:-----:|:-----:|:-----:|
> | [1] CMI         | 62.67 | 56.71 | 53.66 | 49.71 | 44.92 |
> | [2] ShipNet     | 64.68 | 58.74 | 57.93 | 53.93 | 50.71 |
> | [3] NAYER       | 67.82 | 62.56 | 59.52 | 55.79 | 52.69 |
> | DDIS (Ours)     | 69.93 | 66.18 | 63.72 | 59.14 | 55.31 |
>
> **W1)** Previous DFIS methods like DeepInversion lack strong image priors and rely on weak constraints (e.g., total variation), leading to unrealistic artifacts or samples far from the true data manifold (see Figs. 2–5). They also optimize in high-dimensional pixel space (e.g., B×3×256×256) from random noise, making the process unstable and inefficient. In contrast, our method is the first to leverage strong priors from text-to-image (T2I) diffusion models, guiding synthesis toward realistic images and constraining the search space. Moreover, we operate in a compact embedding space (e.g., 1×784), which simplifies optimization. These improvements allow our method to more accurately match the original training distribution and address key limitations of prior work.
>
> **W2,W6)**(Please see this link: https://drive.google.com/file/d/1S2Qt0XSgMH2pOzdzsNekhlyP-bLBY5kt/view?usp=sharing)
> We compare DDIS with methods [1–3] on synthetic data quality and KD performance using CIFAR-10/100, as [1–3] don’t support large-scale, multi-domain synthesis. All methods use ResNet-34 to generate 50K images per dataset. We evaluate IS, FID, precision, and recall (see Fig. 1–2), and Table 1 shows DDIS outperforms all baselines with more realistic images.
> For data-free knowledge distillation (DFKD), we use ResNet-34 as teacher and ResNet-18 as student. While student accuracy is slightly lower than KD-focused methods using special losses, our goal is to match real data distribution, not optimize for KD.
>
> To show generality, we apply DDIS to data-free pruning. Table 2 reports top-1 accuracy of pruned ResNet-34 (ratios 0.5–0.9) fine-tuned on our synthetic data. DDIS consistently achieves the best results, proving its broad applicability.
>
> **W3)** NaturalInversion (NI) focuses on small-scale datasets like CIFAR-10/100 and faces scalability issues due to quality and compute limits. It shifts optimization to parameter space using both a main and sub-generator (FTP), increasing trainable parameters. Scaling to larger datasets requires bigger generators, expanding the search space and computational cost, making large-scale synthesis infeasible. For fair comparison, we apply our method to CIFAR-10/100 with pretrained ResNet-34, generating 50k samples. As shown in Table 1, our method outperforms NI in IS, FID, precision, and recall, better aligning with the original data distribution. These results demonstrate our method’s scalability and effectiveness, and we will include them in the revised paper to strengthen our evaluation.
>
> **W4)** Algorithm 1 outlines Domain Alignment Guidance (DAG) used during diffusion sampling. The noisy latent $z_t$​ is decoded by $D$ into $x_t$​, which is input to classifier $f$ to compute BN loss. The gradient of this loss guides $z_t$​, yielding z_t^{\tilde} (line 6). This guided latent is then used for classifier-free guidance (line 7) and for predicting the next-step latent (line 8), following standard diffusion sampling. This process repeats until z_0^{\tilde} is obtained. The full pipeline with DAG is detailed in Algorithm 2.
>
> **W5)** As detailed in **W1)** to Reviewer #e4Ls and Appendix A.3–A.4.

---

### Decision · Program_Chairs · 2025-05-01

**Decision:**

Accept (poster)

**Comment:**

The paper presents a method for DataFree Image Synthesis using text-to-image diffusion models. reviewers liked the idea and were mostly happy with the experiments and claifications from the authors after the rebuttal. The AC recommends acceptance